# Active Vision Reinforcement Learning under Limited Visual Observability

**Jinghuan Shang**    **Michael S. Ryoo**
Department of Computer Science, Stony Brook University
{jishang, mryoo}@cs.stonybrook.edu

## Abstract

In this work, we investigate Active Vision Reinforcement Learning (ActiveVision-RL), where an embodied agent simultaneously learns action policy for the task while also controlling its visual observations in partially observable environments. We denote the former as *motor policy* and the latter as *sensory policy*. For example, humans solve real world tasks by hand manipulation (motor policy) together with eye movements (sensory policy). ActiveVision-RL poses challenges on coordinating two policies given their mutual influence. We propose SUGARL, Sensorimotor Understanding Guided Active Reinforcement Learning, a framework that models motor and sensory policies separately, but jointly learns them using with an intrinsic sensorimotor reward. This learnable reward is assigned by sensorimotor reward module, incentivizes the sensory policy to select observations that are optimal to infer its own motor action, inspired by the sensorimotor stage of humans. Through a series of experiments, we show the effectiveness of our method across a range of observability conditions and its adaptability to existed RL algorithms. The sensory policies learned through our method are observed to exhibit effective active vision strategies.

## 1   Introduction

Although Reinforcement Learning (RL) has demonstrated success across challenging tasks and games in both simulated and real environments [2, 9, 11, 89, 97], the observation spaces for visual RL tasks are typically predefined to offer the most advantageous views based on prior knowledge and can not be actively adjusted by the agent itself. For instance, table-top robot manipulators often utilize a fixed overhead camera view [51]. While such fixed viewpoints can potentially stabilize the training of an image feature encoder [14], this form of perception is different from humans who actively adjust their perception system to finish the task, e.g. eye movements [24]. The absence of active visual perception poses challenges on learning in highly dynamic environments [59, 61], open-world tasks [30] and partially observable environments with occlusions, limited field-of-views, or multiple view angles [40, 83].

We study Active Reinforcement Learning (Active-RL) [102], the RL process that allows the embodied agent to actively acquire new perceptual information in contrast to the standard RL, where the new information could be reward signals [3, 23, 27, 50, 56, 62], visual observations [32, 33], and other forms. Specifically, we are interested in visual Active-RL tasks, i.e. ActiveVision-RL, that an agent controls its own views of visual observation, in an environment with limited visual observability [33]. Therefore, the goal of ActiveVision-RL is to learn two policies that still maximize the task return: the *motor policy* to finish the task and the *sensory policy* to control the observation.

---

Our project page, code, and library are available at this link

37th Conference on Neural Information Processing Systems (NeurIPS 2023).

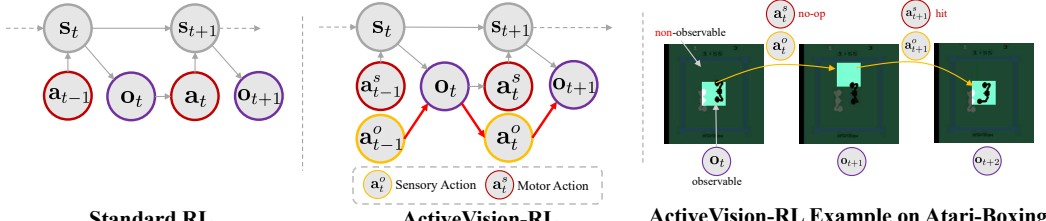

**Standard RL**  **ActiveVision-RL**  **ActiveVision-RL Example on Atari-Boxing**

Figure 1: ActiveVision-RL with limited visual observability in comparison with standard RL, with exemplary process in Atari game *Boxing*. Red arrows stand for additional relationships considered in ActiveVision-RL. In the example on the right, the highlighted regions are the actual observations visible to the agent at each step. The rest of the pixels are not visible to the agent.

ActiveVision-RL tasks present a considerable challenge due to the coordination between motor and sensory policies, given their mutual influence [102]. The motor policy requires clear visual observation for decision-making, while the sensory policy should adapt accordingly to the motor action. Depending on the sensory policy, transitioning to a new view could either aid or hinder the motor policy learning [14, 40, 102]. One notable impediment is the perceptual aliasing mentioned by Whitehead and Ballard [102]. An optimal strategy for sensory policy should be incorporating crucial visual information while eliminating any distractions. In the real world, humans disentangle their sensory actions, such as eye movements, from their motor actions, such as manipulation, and subsequently learn to coordinate them [59, 61]. Despite being modeled separately, these two action policies and the coordination can be learned jointly through the interaction during sensorimotor stage [25, 71, 72, 103].

Taking inspiration from human capabilities, we propose SUGARL: Sensorimotor Understanding Guided Active Reinforcement Learning, an Active-RL framework designed to jointly learn sensory and motor policies by maximizing extra intrinsic sensorimotor reward together with environmental reward. We model the ActiveVision-RL agent with separate sensory and motor policies by extending existing RL algorithms with two policy/value branches. Inspired by sensorimotor stage [71, 72, 103], we use the intrinsic sensorimotor reward to guide the joint learning of two policies, imposing penalties on the agent for selecting sub-optimal observations. We use a learned sensorimotor reward module to assign the intrinsic reward. The module is trained using inverse dynamics prediction task [52, 95], with the same experiences as policy learning without additional data or pre-training.

In our experiments, we use modified Atari [9] and DeepMind Control suite (DMC) [97] with limited observability to comprehensively evaluate our proposed method. We also test on Robosuite tasks to demonstrate the effectiveness of active agent in 3D manipulation. Through the challenging benchmarks, we experimentally show that SUGARL is an effective and generic approach for Active-RL with minimum modification on top of existed RL algorithms. The learned sensory policy also exhibit active vision skills by analogy with humans' fixation and tracking.

## 2  Active Vision Reinforcement Learning Settings

Consider a vanilla RL setting based on a Markov Decision Process (MDP) described by $(\mathcal{S}, \mathcal{A}, r, P, \gamma)$, where $\mathcal{S}$ is the state space, $\mathcal{A}$ is the action space, $r$ is the reward function, $P$ describes state transition which is unknown and $\gamma$ is the discount factor. In this work, we study ActiveVision-RL under limited visual observability, described by $(\mathcal{S}, \mathcal{O}, \mathcal{A}^s, \mathcal{A}^o, r, P, \gamma)$, as shown in Figure 1. $\mathcal{O}$ is the actual partial observation space the agent perceives. In particular, we are interested in visual tasks, so each observation $\mathbf{o}$ is an image contains partial information of an environmental state $\mathbf{s}$, like an image crop in 2D space or a photo from a viewpoint in 3D space. To emulate the human ability, there are two action spaces for the agent in Active-RL formulation. $\mathcal{A}^s$ is the motor action space that causes state change $p(\mathbf{s}'|\mathbf{s}, \mathbf{a}^s)$. $\mathcal{A}^o$ is the sensory action space that only changes the observation of an environmental state $p(\mathbf{o}|\mathbf{s}, \mathbf{a}^o)$. In this setting, the agent needs to take $(\mathbf{a}^s, \mathbf{a}^o)$ for each step, based on observation(s) only. An example is shown in Figure 1.

Our goal is to learn the motor and sensory action policies $(\pi^s, \pi^o)$ that still maximize the return $\sum r_t$. Note that the agent is never exposed to the full environmental state $\mathbf{s}$. Both policies are completely based on the partial observations: $\mathbf{a}^s = \pi^s(\cdot|\mathbf{o})$, $\mathbf{a}^o = \pi^o(\cdot|\mathbf{o})$. Therefore the overall policy learning is challenging due to the limited information per step and the non-stationary observations.

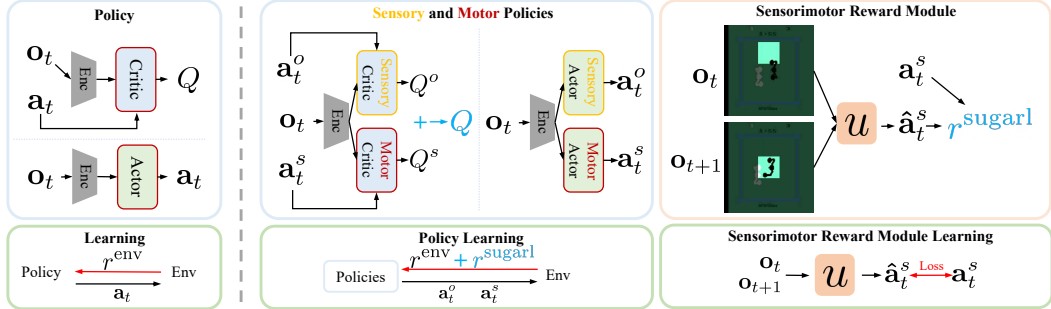

Figure 2: Overview of SUGARL and the comparison with original RL algorithm formulation. SUGARL introduces an extra sensory policy head, and jointly learns two policies together with the extra sensorimotor reward. We use the formulation of SAC [36] as an example. We introduce sensorimotor reward module to assign the reward. The reward indicates the quality of the sensory policy through the prediction task. The sensorimotor reward module is trained independently to the policies by action prediction error.

## 3 SUGARL: Sensorimotor Understanding Guided Active-RL

### 3.1 Active-RL Algorithm with Sensorimotor Understanding

We implement Active-RL algorithms based on the normal vision-based RL algorithms with simple modifications regarding separated motor and sensory policies $(\pi^s, \pi^o)$, and the sensorimotor reward $r^{\text{sugarl}}$. We use DQN [65] and SAC [36] as examples to show the modifications are generally applicable. The example diagram of SAC is shown in Figure 2. We first introduce the policy then describe the sensorimotor reward in Section 3.2

**Network Architecture** The architectural modification is branching an additional head for sensory policy, and both policies share a visual encoder stem. For DQN [65], two heads output $Q^s, Q^o$ for each policy respectively. This allows the algorithm to select $\mathbf{a}^s = \arg\max_{\mathbf{a}^s} Q^s$ and $\mathbf{a}^o = \arg\max_{\mathbf{a}^o} Q^o$ for each step. Similarly for SAC [36], the value and actor networks also have two separate heads for motor and sensory policies. The example in the form of SAC is in Figure 2.

**Joint Learning of Motor and Sensory Policies** Though two types of actions are individually selected or sampled from network outputs, we find that the joint learning of two policies benefits the whole learning [41, 111]. The joint learning here means both policies are trained using a shared reward function. Otherwise, the sensory policy usually fails to learn with intrinsic reward signal only. Below we give the formal losses for DQN and SAC.

For DQN, we take the sum of Q-values from two policies $Q = Q^s + Q^o$ and train both heads jointly. The loss is the following where we indicate our modifications by blue:

$$\mathcal{L}_i^Q(\theta_i) = \mathbb{E}_{(\mathbf{o}_t, \mathbf{a}_t^s, \mathbf{a}_t^o) \sim \mathcal{D}} \left[ \left( y_i - \left( Q_{\theta_i}^s(\mathbf{o}_t, \mathbf{a}_t^s) + Q_{\theta_i}^o(\mathbf{o}_t, \mathbf{a}_t^o) \right) \right)^2 \right]$$

$$y_i = \mathbb{E}_{\mathbf{o}_{t+1}} \left[ r_t^{\text{env}} + \beta r_t^{\text{sugarl}} + \gamma \left( \max_{\mathbf{a}_{t+1}^s} Q_{\theta_{i-1}}^s(\mathbf{o}_{t+1}, \mathbf{a}_{t+1}^s) + \max_{\mathbf{a}_{t+1}^o} Q_{\theta_{i-1}}^o(\mathbf{o}_{t+1}, \mathbf{a}_{t+1}^o) \right) \right],$$

where $L^Q$ is the loss for Q-networks, $\theta$ stands for the parameters of both heads and the encoder stem, $i$ is the iteration, and $\mathcal{D}$ is the replay buffer. $\beta r_t^{\text{sugarl}}$ is the extra sensorimotor reward with balancing scale which will be described in Section 3.2.

For SAC, we do the joint learning similarly. The soft-Q loss is in the similar form of above DQN which is omitted for simplicity. The soft-value loss $L^V$ and is

$$\mathcal{L}^V(\psi) = \mathbb{E}_{\mathbf{o}_t \sim \mathcal{D}} \left[ \frac{1}{2} \left( V_\psi^s(o_t) + V_\psi^o(o_t) - \right. \right.$$

$$\left. \left. \mathbb{E}_{\mathbf{a}_t^s \sim \pi_\phi^s, \mathbf{a}_t^o \sim \pi_\phi^o} \left[ Q_\psi^s(\mathbf{o}_t, \mathbf{a}_t^s) + Q_\psi^o(\mathbf{o}_t, \mathbf{a}_t^o) - \log \pi_\phi^s(\mathbf{a}_t^s | o_t) - \log \pi_\phi^o(\mathbf{a}_t^o | o_t) \right] \right)^2 \right],$$

and the actor loss $\mathcal{L}^\pi$ is

$$\mathcal{L}^\pi(\phi) = \mathbb{E}_{\mathbf{o}_t \sim \mathcal{D}} \left[ \log \pi_\phi^s(\mathbf{a}_t^s | \mathbf{o}_t) + \log \pi_\phi^o(\mathbf{a}_t^o | \mathbf{o}_t) - Q_\psi^s(\mathbf{o}_t, \mathbf{a}_t^s) - Q_\psi^o(\mathbf{o}_t, \mathbf{a}_t^o) \right],$$

where $\psi, \phi$ are parameters for the critic and the actor respectively, and reparameterization is omitted for clearness.

## 3.2 Sensorimotor Reward

The motor and sensory policies are jointly trained using a shared reward function, which is the combination of environmental reward and our sensorimotor reward. We first introduce the assignment of sensorimotor reward and then describe the combination of two rewards.

The sensorimotor reward is assigned by the sensorimotor reward module $u_\xi(\cdot)$. The module is trained to have the sensorimotor understanding, and is used to indicate the goodness of the sensory policy, tacking inspiration of human sensorimotor learning [72]. The way we obtain such reward module is similar to learning an inverse dynamics model [95]. Given a transition $(\mathbf{o}_t, \mathbf{a}_t^s, \mathbf{o}_{t+1})$, the module predicts the motor action $\mathbf{a}^s$ only, based on an observation transition tuple $(\mathbf{o}_t, \mathbf{o}_{t+1})$. When the module is (nearly) fully trained, the higher prediction error indicates the worse quality of visual observations. For example, if the agent is absent from observations, it is hard to infer the motor action. Such sub-optimal observations also confuse agent's motor policy. Since the sensory policy selects those visual observations, the quality of visual observations is tied to the sensory policy. As a result, we can employ the negative error of action prediction as the sensorimotor reward:

$$r_t^{\text{sugarl}} = -\left(1 - p(\mathbf{a}_t^s | \mathbf{o}_t, \mathbf{o}_{t+1}; u_\xi)\right). \tag{1}$$

This non-positive intrinsic reward penalizes the sensory policy for selecting sub-optimal views that do not contribute to the accuracy of action prediction and confuse motor policy learning. In Section 5.4 we show that naive positive rewarding does not guide the policy well. Note that the reward is less noisy when the module is fully trained. However, it is not harmful for being noisy at the early stage of training, as the noisy signal may encourage the exploration of policies. We use the sensorimotor reward though the whole learning.

The sensorimotor reward module is implemented by an independent neural network. The loss is a simple prediction error:

$$\mathcal{L}^u(\xi) = \mathbb{E}_{\mathbf{o}_t, \mathbf{o}_{t+1}, \mathbf{a}_t^s \sim \mathcal{D}} \left[ \text{Error}\left(\mathbf{a}_t^s - u_\xi(\mathbf{o}_t, \mathbf{o}_{t+1})\right) \right], \tag{2}$$

where the Error$(\cdot)$ can be a cross-entropy loss for discrete action space or L2 loss for continuous action space. Though being implemented and optimized separately, the sensorimotor reward module uses the same experience data as policy learning, with no extra data or prior knowledge introduced.

**Combining Sensorimotor Reward and Balancing** The sensorimotor reward $r^{\text{sugarl}}$ is added densely on a per-step basis, on top of the environmental reward $r^{\text{env}}$ in a balanced form:

$$r_t = r_t^{\text{env}} + \beta r_t^{\text{sugarl}}, \tag{3}$$

where $\beta$ is the balance parameter varies across environments. The reward balance is very important to make both motor and sensory policies work as expected, without heavy bias towards one side [28], which will be discussed in our ablation study in Section 5.4. Following the studies in learning with intrinsic rewards and rewards of many magnitudes [20, 38, 75, 96, 99], we empirically set $\beta = \mathbb{E}_\tau[\sum_{t=1}^T r_t^{\text{env}}/T]$, which is the average environmental return normalized by the length of the trajectory. We get these referenced return data from the baseline agents trained on normal, fully observable environments, or from the maximum possible environmental return of one episode.

## 3.3 Persistence-of-Vision Memory

To address ActiveVision-RL more effectively, we introduce a Persistence-of-Vision Memory (PVM) to spatio-temporally combine multiple recent partial observations into one, mimicking the nature

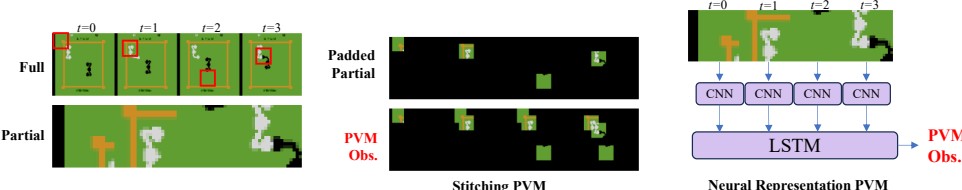

Figure 3: Examples for different instantiations of PVM with $B = 3$.

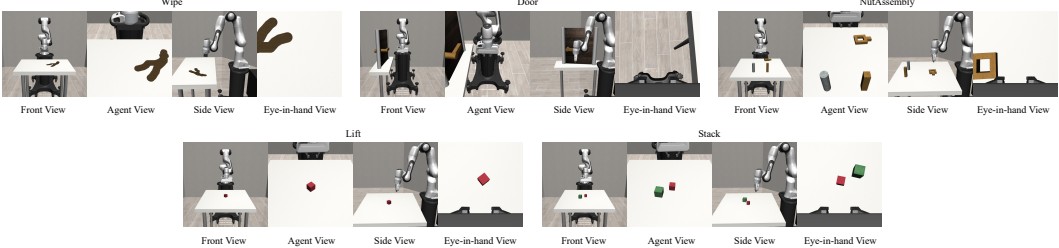

Figure 4: Five selected Robosuite tasks with examples on four hand-coded views.

of human eyes and the memory. PVM aims to expand the effective observable area, even though some visual observations may become outdated or be superseded by more recent observations. PVM stores the observations from $B$ past steps in a buffer and combines them into a single PVM observation according to there spatial positions. This PVM observation subsequently replaces the original observation at each step:

$$\text{PVM}(\mathbf{o}_t) = f(\mathbf{o}_{t-B+1}, \ldots, \mathbf{o}_t),$$

where $f(\cdot)$ is a combination operation. In the context of ActiveVision-RL, it is reasonable to assume that the agent possesses knowledge of its focus point, as it maintains the control over the view. Therefore, the viewpoint or position information can be used. In our implementations of $f(\cdot)$ shown in Figure 3, we show a 2D case of PVM using stitching, and a PVM using LSTM which can be used in both 2D and 3D environments. In stitching PVM, the partial observations are combined like a Jiasaw puzzle. It is worth noting that combination function $f$ can be implemented by other pooling operations, sequence representations [15, 44, 85, 86], neural memories [37, 39, 76, 78], or neural 3D representations [74, 107], depending on the input modalities, tasks, and approaches.

## 4 Environments and Settings

### 4.1 Active-Gym: An Environment for ActiveVision-RL

We present Active-Gym, an open-sourced customized environment wrapper designed to transform RL environments into ActiveVision-RL constructs. Our library currently supports active vision agent on Robosuite [114], a robot manipulation environment in 3D, as well as Atari games [9] and the DeepMind Control Suite (DMC) [97] offering 2D active vision cases. These 2D environments were chosen due to the availability of full observations for establishing baselines and upper bounds, and the availability to manipulate observability for systematic study.

### 4.2 Robosuite Settings

**Observation Space**  In the Robosuite environment, the robot controls a movable camera. The image captured by that camera is a partial observation of the 3D space.

**Action Spaces**  Each step in Active-Gym requires motor and sensory actions $(\mathbf{a}^s, \mathbf{a}^o)$. The motor action space $\mathcal{A}^s$ is the same as the base environment. The sensory action space $\mathcal{A}^o$ is a 5-DoF control: relative (x, y, z, yaw, pitch). The maximum linear and angular velocities are constrained to 0.01/step and 5 degrees/step, respectively.

### 4.3 2D Benchmark Settings

**Observation Space**  In the 2D cases of Active-Gym, given a full observation $\mathbf{O}$ with dimensions $(H, W)$, only a crop of it is given to the agent's input. Examples are highlighted in red boxes in Figure 5. The sensory action decides an observable area by a location $(x, y)$, corresponding to the top-left corner of the bounding box, and the size of the bounding box $(h, w)$. The pixels within the observable area becomes the foveal observation, defined as $\mathbf{o}^f = \mathbf{o}^c = \mathbf{O}[x : x + h, y : y + w]$. Optionally, the foveal observation can be interpolated to other resolutions $\mathbf{o}^f = \text{Interp}(\mathbf{o}^c; (h, w) \to (r_h^f, r_w^f))$, where $(r_h^f, r_w^f)$ is the foveal resolution. This design allows for flexibility in altering the observable area size while keeping the effective foveal resolution constant. Typically we set $(r_h^f, r_w^f) = (h, w)$ and fixed them during a task. The peripheral observation can be optionally provided as well, obtained by interpolating the non-foveal part $\mathbf{o}^p = \text{Interp}(\mathbf{O} \setminus \mathbf{o}^c; (H, W) \to (r_h^p, r_w^p))$, where $(r_h^p, r_w^p)$ is the peripheral resolution. The examples are at the even columns of Figure 5. If the peripheral observation is not provided, $\mathbf{o}^p = 0$.

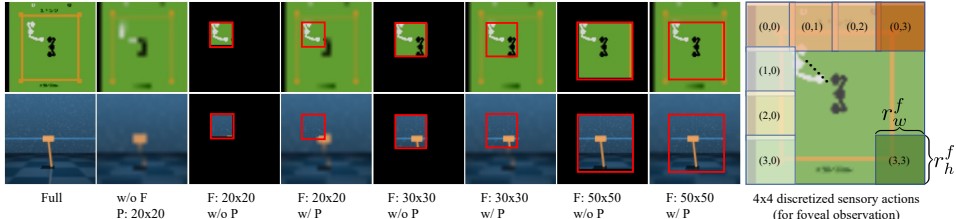

| Full | w/o F P: 20x20 | F: 20x20 w/ P | F: 20x20 w/o P | F: 30x30 w/o P | F: 30x30 w/ P | F: 50x50 w/o P | F: 50x50 w/ P | 4x4 discretized sensory actions (for foveal observation) |

Figure 5: Left: observations from Active-Gym with different foveal resolutions (F) and peripheral settings (P) in Atari and DMC. The red bounding boxes show the observable area (foveal) for clarity. Right: 4x4 discrete sensory action options in our experiments.

Table 1: Results on Robosuite. We report the IQM of raw rewards from 30 evaluations. Highlighted task names are harder tasks. **Bold** numbers are the best scores of each task and underscored numbers are the second best.

| Approach | Wipe | Door | NutAssemblySquare | Lift | Stack |
|---|---|---|---|---|---|
| SUGARL-DrQ (Stacking PVM) | 56.0 | 274.8 | 78.0 | 79.2 | 12.7 |
| SUGARL-DrQ (LSTM PVM) | 58.5 | 266.9 | **108.6** | 88.8 | 31.5 |
| SUGARL-DrQ (3D Transformation+LSTM PVM) | **74.1** | **291.0** | 65.2 | 87.5 | 32.4 |
| SUGARL-DrQ w/o Joint Learning | 43.6 | 175.4 | 58.0 | 107.2 | 12.0 |
| SUGARL-DrQ w/o PVM | 52.8 | 243.3 | 37.9 | 55.6 | 7.7 |
| Single Policy | 12.4 | 22.8 | 8.42 | 10.7 | 0.53 |
| DrQ w/ Object Detection (DETR) | 15.2 | 43.1 | 54.8 | 15.4 | 7.5 |
| DrQ w/ End-to-End Attention | 14.2 | 141.4 | 28.5 | 33.0 | 13.6 |
| Eye-in-hand View (hand-coded, moving camera) | 16.1 | 114.6 | 102.9 | **233.9** | **73.0** |
| Front View (hand-coded, fixed camera) | 49.4 | 240.6 | 39.6 | 69.0 | 13.8 |
| Agent View (hand-coded, fixed camera) | 12.7 | 190.3 | 49.9 | 122.6 | 14.7 |
| Side View (hand-coded, fixed camera) | 25.9 | 136.2 | 34.5 | 56.6 | 12.8 |

**Action Spaces** The sensory action space $\mathcal{A}^o$ includes all the possible (pixel) locations on the full observation, but can be further formulated to either continuous or discrete spaces according to specific task designs. In our experiments, we simplify the space by a 4x4 discrete grid-like anchors for $(x, y)$ (Figure 5 right). Each anchor corresponds to the top-left corner of the observable area. The sensory policy chooses to place the observable area among one of 16 anchors (**absolute** control), or moves it from one to the four neighbor locations (**relative** control).

## 4.4 Learning Settings

In our study, we primarily use DQN [65] and SAC [36] as the backbone algorithms of SUGARL to address tasks with discrete action spaces (Atari), and use DrQv2 [108] for continuous action spaces (DMC and Robosuite). All the visual encoders are standardized as the convolutional networks utilized in DQN [65]. To keep the network same, we resize all inputs to 84x84. For the sensorimotor understanding model, we employ the similar visual encoder architecture with a linear head to predict $\mathbf{a}_t^s$. Each agent is trained with one million transitions for each of the 26 Atari games, or trained with 0.1 million transitions for each of the 6 DMC tasks and 5 Robosuite tasks. The 26 games are selected following Atari-100k benchmark [47]. We report the results using Interquartile Mean (IQM), with the scores normalized by the IQM of the base DQN agent under full observation (except Robosuite), averaged across five seeds (three for Robosuite) and all games/tasks per benchmark. Details on architectures and hyperparameters can be found in the Appendix.

## 5 Results

### 5.1 Robosuite Results

We selected five of available tasks in Robosuite [114], namely block lifting (Lift), block stacking (Stack), nut assembling (NutAssembleSquare), door opening (Door), wiping the table (Wipe). The first two are easier compared to the later three. Example observations are available in Figure 4. We compare against a straightforward baseline that a single policy is learned to govern both motor and sensory actions. We also compare to baselines including RL with object detection (a replication of Cheng et al. [17]), learned attention [93], and standard RL with hand-coded views. Results are in 1. We confirm that our SUGARL works outperforms baselines all the time, and also outperforms

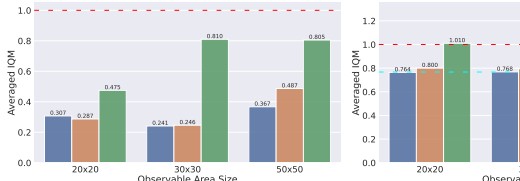
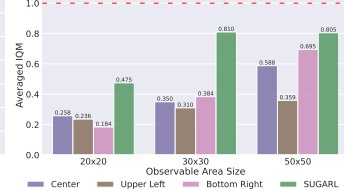

| (a) Without peripheral observation | (b) With peripheral observation | (c) Static policies comparison |

Figure 6: Results with different observation size and peripheral observation settings. The green bars are results from SUGARL. The red dashed lines stand for the DQN baseline trained on full observations using the same amount of data. We compare SUGARL against two dynamic views: Random View and Raster Scanning, and three static view baselines. In (b) with peripheral observation, we compare a baseline using peripheral observation only indicated by the cyan line.

the hand-coded views most of the time. Specifically, for the harder tasks including Wipe, Door, and NutAssemblySquare, SUGARL gets the best scores.

**Designs of PVM** We compare different instantiations of proposed PVMs including: Stacking: naively stacking multiple frames; LSTM: Each image is first encoded by CNN and fed into LSTM; 3D Transformation + LSTM: we use camera parameters to align pixels from different images to the current camera frame. Then an LSTM encodes the images after going through CNN. We find that 3D Transformation + LSTM works the best, because it tackles spatial aligning and temporal merging together. LSTM also works well in general.

## 5.2 2D Benchmark Results

We evaluate the policy learning on Atari under two primary visual settings: **with** and **without peripheral observation**. In each visual setting we explore three sizes of observable area (set equivalent to foveal resolution): **20x20**, **30x30**, and **50x50**. In with peripheral observation setting, the peripheral resolution is set to 20x20 for all tasks. We use DQN [65]-based SUGARL (SUGARL-DQN) and compare it against two variants by replacing the learnable sensory policy in SUGARL with: **Random View** and **Raster Scanning**. Random View always uniformly samples from all possible crops. Raster Scanning uses a pre-designed sensory policy that chooses observable areas from left to right and top to down sequentially. Raster Scanning yields relatively stable observation patterns and provides maximum information under PVM. We also provide a DQN baseline trained on full observations (84x84) as a soft oracle. In with peripheral observation settings, another baseline trained on peripheral observation only (20x20) is compared as a soft lower bound.

In Figure 6a and 6b, we find that SUGARL performs the best in all settings, showing that SUGARL learns effective sensory and motor policies jointly. More importantly, SUGARL with peripheral observation achieves higher overall scores (+0.01~0.2) than the full observation baselines. In details, SUGARL gains higher scores than the full observation baseline in 13 out of 26 games with 50x50 foveal resolution (details are available in the Appendix). This finding suggests the untapped potential of ActiveVision-RL agents which leverage partial observations better than full observations. By actively selecting views, the agent can filter out extraneous information and concentrate on task-centric information.

**Compare against Static Sensory Policies** We also examine baselines with static sensory policies, which consistently select one region to observe throughout all steps. The advantage of this type baseline lies in the stability of observation. We select 3 regions: **Center**, **Upper Left**, and **Bottom Right**, and compare them against SUGARL in environment w/o peripheral observation. As shown in Figure 6c, we observe that SUGARL still surpasses all three static policies. The performance gaps between SUGARL and Center and Bottom Right are relatively small when the observation size is larger (50x50), as the most valuable information is typically found at these locations in Atari

Table 2: Evaluation results on different conditions and algorithm backbones

| (a) Action modeling | | | |
|---|---|---|---|
| Model | 20 | 30 | 50 |
| SUGARL (abs) | **0.475** | **0.810** | 0.805 |
| SUGARL (rel) | 0.367 | 0.745 | **0.945** |
| Single Policy | 0.132 | 0.222 | 0.171 |

| (b) Train more steps | | | | |
|---|---|---|---|---|
| Steps | Model | 20 | 30 | 50 |
| 1M | SUGARL | **0.475** | **0.810** | 0.805 |
| | Single Policy | 0.132 | 0.222 | 0.171 |
| 5M | SUGARL | 1.170 | 1.121 | 1.553 |
| | Single Policy | 0.332 | 0.640 | 1.145 |

| (c) SUGARL with SAC | | | |
|---|---|---|---|
| Model | 20 | 30 | 50 |
| SUGARL | **0.424** | **0.730** | **0.785** |
| SUGARL w/o $r^{sugarl}$ | 0.300 | 0.307 | 0.504 |
| SAC-raster scanning | 0.117 | 0.195 | 0.136 |
| SAC-random view | 0.155 | 0.104 | 0.134 |

| (d) Different PVMs | | | |
|---|---|---|---|
| Model | 20 | 30 | 50 |
| Stitching PVM | **0.475** | **0.815** | **0.810** |
| LSTM PVM | 0.397 | 0.448 | 0.470 |

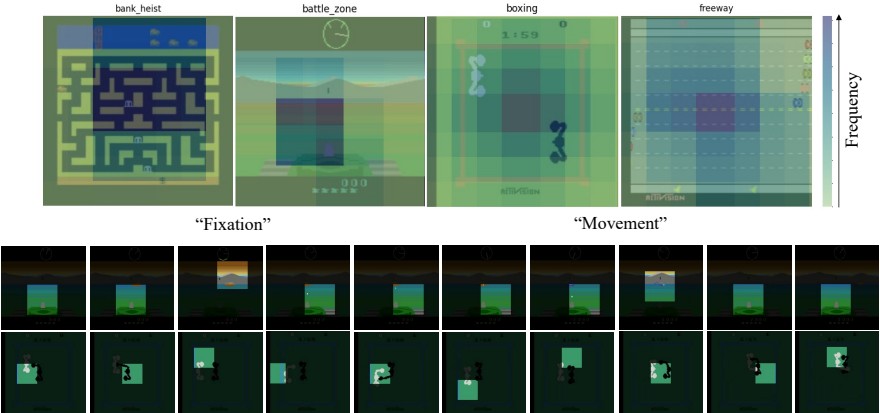

Figure 7: Examples of learned sensory policies from SUGARL-DQN. Top: frequency heat maps of pixels being observed. Bottom: sequences of observations showing fixation-like and tracking-like behaviors. More videos are available at this link

environments. Static observation is therefore quite beneficial for decision-making. However, as the observation size decreases, an active policy becomes essential for identifying optimal observations, leading SUGARL to outperform others by a significant margin.

**Sensory-Motor Action Spaces Modeling**   SUGARL models the sensory and motor action spaces separately inspired by the human abilities. An alternative approach is jointly modeling two action spaces $\mathcal{A} = \mathcal{A}^s \times \mathcal{A}^o$ and learning one single action policy, with environmental reward only (referred as **Single Policy**). We compare Single Policy and display the results in Table 2a, which reveal that modeling one large action space fails to learn the policy properly due to the expanded action space. Additionally, we examine two types of sensory action spaces: **absolute** (abs) and **relative** (rel). Absolute modeling allows the view to jump across the entire space while relative modeling restricts the view to moving only to the neighboring options at each step. Results in Table 2a indicate that the absolute modeling performs better in smaller observable regions, as it can select the desired view more quickly than relative modeling. In contrast, relative modeling demonstrates better performance in larger observable region setting as it produces more stable observations across steps.

**Training More Steps**   We subsequently explore SUGARL performance when trained with more environmental steps, comparing outcomes between 1M and 5M steps. Results in Table 2b confirm that SUGARL continuous to improve with more steps and consistently outperforms the single policy baseline, suggesting that it does not merely stagnate at a trivial policy. However, due to the challenges posed by limited observability, the learning still proceeds slower than for the agent with full observations, which achieves the score 4.106 at 5M steps.

**Generalization of SUGARL**   We apply SUGARL to Soft Actor-Critic (SAC) [36] and evaluate its performance on environments without peripheral observation. As before, we compare it with Random View, Raster Scanning, and SUGARL without intrinsic reward. According to Table 2c, we find that SUGARL-SAC outperforms the naive version without the guidance of $r^{\text{sugarl}}$, further emphasizing the significance of our method. Moreover, SUGARL-SAC also surpasses the random view and raster scanning policies. However, when employing max-entropy-based methods like SAC, it is necessary to reduce the weight of the entropy term to ensure consistency in the sensory policy. We will further discuss it in Section 5.3 based on the analysis of the learned sensory policy behavior.

**DMC Tasks**   We also evaluate SUGARL on DMC [97] tasks based on DrQ [108], an improved version of DDPG [100]. We use six environments in DMC [97]: *ball_in_cup-catch*, *cartpole-swingup*, *cheetah-run*, *dog-fetch*, *fish-swim*, and *walker-walk*. Three foveal observation resolutions (20x20, 30x30, and 50x50) without peripheral observation are explored. We use relative control for sensory action and decide the discrete sensory options by thresholding the continuous output. We compare our approach with baselines with joint modeling of sensory and motor action spaces (Single Policy), Raster Scanning, and Random View. Results are shown in Table 3. SUGARL outperforms all baselines in three foveal settings in DMC, showing it is also applicable to continuous control tasks.

Table 3: SUGARL on DMC

| Model | 20 | 30 | 50 |
|---|---|---|---|
| SUGARL-DrQ | **0.686** | **0.717** | **1.052** |
| DrQ-Single Policy | 0.540 | 0.570 | 0.776 |
| DrQ-Raster Scanning | 0.609 | 0.566 | 0.913 |
| DrQ-Random View | 0.569 | 0.591 | 0.768 |
| SUGARL-DrQ w/o PVM | 0.672 | 0.620 | 0.930 |
| SUGARL w/o Joint Learning | 0.377 | 0.446 | 0.355 |

## 5.3 Sensory Policy Analysis

**Sensory Policy Patterns** In Figure 7, we present examples of learned sensory policies from SUGARL-DQN in four Atari games, in settings w/o peripheral observations. These policies are visualized as heat maps based on the frequency of observed pixels. We discover that the sensory policies learn both **fixation** and **movement** (similar to tracking) behaviours [12, 110] depending on the specific task requirements. In the first two examples, the sensory policy tends to concentrate on fixed regions. In *battle_zone*, the policy learns to focus on the regions where enemies appear and the fixed front sight needed for accurate firing. By contrast, in highly dynamic environments like *boxing* and *freeway*, the sensory policies tend to observe broader areas in order to get timely observations. Though not being a perfect tracker, the sensory policy learns to track the agent or the object of interest in these two environments, demonstrating the learned capability akin to humans' Smooth Pursuit Movements. Recorded videos for entire episodes are available at our project page.

**Sensory Policy Distribution** We quantitatively assess the distributions of learned sensory policies. There are 16 sensory actions, i.e. the observable area options in our setup (Figure 5). We compare the distributions against uniform distribution using KL-divergence, across 26 games x 10 eval runs x 5 seeds. The resulting histogram are shown in Figure 8. We observe that the learned policies consistently deviate from the uniform distribution, suggesting that sensory policies prefer specific regions in general. The high peak at the high KL end supports the "fixation" behavior identified in the previous analysis. As the observation size increases, the divergence distribution shifts towards smaller KL end, while remaining $> 0.5$ for all policies. This trend indicates that with larger observable sizes, the policy does not need to adjust its attention frequently, corroborating the benefit of using relative control shown in Table 2a.

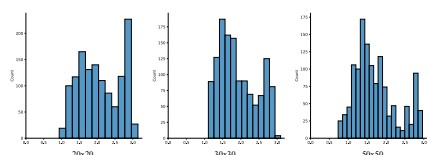

Figure 8: KL divergence distributions of learned sensory policies.

**Pitfalls on Max-Entropy Based Methods** In the previous analysis, we both qualitatively and quantitatively demonstrate that sensory policies are not uniformly dispersed across the entire sensory action space. This observation implies that *sensory* policy should exercise caution when adopting the max-entropy-based methods. We conduct an experiment on varying the usage of $\alpha$, the entropy coefficient in SAC in all three foveal observation size settings. Results in Table 4 show that simply disabling autotune and setting a small value to the sensory policy's $\alpha$ improves. This finding tells that max-entropy may not be a suitable assumption in modeling sensory policies.

Table 4: Varing $\alpha$ of SUGARL-SAC.

| Model | 20 | 30 | 50 |
|---|---|---|---|
| autotune $\alpha$ | 0.271 | 0.358 | 0.444 |
| fixed-$\alpha = 0.2$ | **0.424** | **0.730** | **0.785** |

## 5.4 Ablation Studies

We conduct ablation studies in the setting without peripheral observation and with 50x50 observation size. Five crucial design components are investigated to validate their effectiveness by incrementally adding or removing them from the full model. The results are presented in Table 5. From the results, we demonstrate the importance of *penalizing* the agent for inaccurate self-understanding predictions rather than rewarding accurate predictions ($r^{\text{sugarl}}(\text{positive}) \to r^{\text{sugarl}}(\text{negative})$). By imposing penalties, the maximum return is bounded by the original maximum possible return per episode, allowing motor and sensory policies to better coordinate each other and achieve optimal task performance. *Reward balance* significantly improves the policy, indicating its effectiveness in coordinating two policies as well. PVM also considerably enhances the algorithm by increasing the effective observable area as expected.

Table 5: Ablation study results based on DQN 50x50 foveal res w/o peripheral observation. The blank fields means there are no such modifications for that model.

| Model | | $r^{\text{sugarl}}$ | Joint Learning | PVM | Reward Balance ($\beta$) | IQM |
|---|---|---|---|---|---|---|
| Random View | | | | ✓ | | 0.367 |
| SUGARL = | Base RL algorithm | | | | | - |
| | +Naive positive intrinsic reward | positive | | | | 0.281 |
| | +Joint learning | positive | ✓ | | | 0.322 |
| | Positive $\to$ negative $r^{\text{sugarl}}$ | negative | ✓ | | | 0.360 |
| | +PVM | negative | ✓ | ✓ | | 0.423 |
| | +Reward Balance | negative | ✓ | ✓ | ✓ | **0.805** |
| SUGARL w/o $r^{\text{sugarl}}$ | | | ✓ | ✓ | | 0.421 |
| SUGARL w/o $r^{\text{sugarl}}$ and w/o PVM | | | ✓ | | | 0.231 |

# 6 Related Work

**Active Learning** is the concept that an agent decides which data are taken into its learning and may ask for external information in comparison to fitting a fixed data distribution [1, 6, 10, 21, 22, 31, 46, 49, 55, 57, 60, 64, 66, 73, 81, 88, 90, 101, 105, 113]. **Active Vision** focuses on continuously acquiring new visual observations that is helpful for the vision task like object classification, recognition and detection [4, 5, 7, 8, 18, 28, 29, 48, 63, 98, 106, 109], segmentation [13, 58, 70], and action recognition [45]. The active vision is usually investigated under a robot vision scenario that a robot moves around in a scene. However, the policy is usually not required to accomplish a task with physical interactions such as manipulating objects compared to reinforcement learning.

**Active Reinforcement Learning** (Active-RL), at a high level, is that the agent is allowed to actively gather new perceptual information of interest simultaneously through an RL task, which can also be called active perception [102]. The extra perceptual information could be reward signal [3, 23, 27, 50, 56, 62], visual observations from new viewpoints [33, 54, 67, 87], other input modalities [16], and language instructions [19, 68, 69, 94]. Though these work may not explicitly use the term Active-RL, we find that they can be uniformly organized in the general Active-RL formulation and we coin the term here. In our work, we study the ActiveVision-RL task in a limited visual observability environment, where at each step the agent is only able to partially observe the environment. The agent should actively seek the optimal observation at each step. Therefore, our setting is more close to [32, 33] and Active Vision problems, unlike research incorporating attention-like inductive bias given a full observation [34, 42, 43, 79, 91, 93, 104]. The ActiveVision-RL agent must learn an observation selection policy, called sensory policy, to effectively choose the optimal partial observation for executing the task-specific policy (motor policy). The unique challenge for ActiveVision-RL is the coordination between sensory and motor policies given there mutual influence. In recent works, the sensory policy can be either trained in the task-agnostic way [33] with enormous exploration data, or trained jointly with the task [32] with naive environmental reward only. In this work we investigate the joint learning case because of the high cost and availability concern of pre-training tasks [33].

**Robot Learning with View Changes** Viewpoint changes and gaps in visual observations are the common challenges for robot learning [40, 53, 77, 84, 92, 112], especially for the embodied agents that uses its first-person view [30, 35, 80]. To address those challenges, previous works proposed to map visual observation from different viewpoints to a common representation space by contrastive encoding [26, 82, 83] or build implicit neural representations [53, 107]. In many first-person view tasks, the viewpoint control is usually modeled together with the motor action like manipulation and movement [30, 80]. In contrast, in our ActiveVision-RL setting, we explore the case where the agent can choose where to observe independently to the motor action inspired by the humans' ability.

# 7 Limitations

In this work, we assume that completely independent sensory and motor actions are present in an embodied agent. But in a real-world case, the movement of the sensors may depend on the motor actions. For example, a fixed camera attached to the end-effector of a robot manipulator, or to a mobility robot. To address the potential dependence and conflicts between two policies in this case, extensions like voting or weighing across two actions to decide the final action may be required. The proposed algorithm also assumes a chance to adjust viewpoints at every step. This could be challenging for applications where the operational or latency costs for adjusting the sensors are high like remote control. To resolve this, additional penalties on sensory action and larger memorization capability are potentially needed. Last, the intrinsic reward currently only considers the accuracy of agent-centric prediction. Other incentives like gathering novel information or prediction accuracy over other objects in the environment can be further explored.

# 8 Conclusion

We present SUGARL, a framework based on existed RL algorithms to jointly learn sensory and motor policies through the ActiveVision-RL task. In SUGARL, an intrinsic reward determined by sensorimotor understanding effectively guides the learning of two policies. Our framework is validated in both 3D and 2D benchmarks with different visual observability settings. Through the analysis on the learned sensory policy, it shows impressive active vision skills by analogy with human's fixation and tracking that benefit the overall policy learning. Our work paves the initial way towards reinforcement learning using active agents for open-world tasks.

## Acknowledgments and Disclosure of Funding

We appreciate the fruitful discussions on the methodology with Xiang Li and Wensheng Cheng, on experimental designs with Kumara Kahatapitiya and Ryan Burgert. We also appreciate the inspiring feedbacks from Kanchana Ranasinghe and Varun Belagali.

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

# A   Appendix

## A.1   Learning Curves

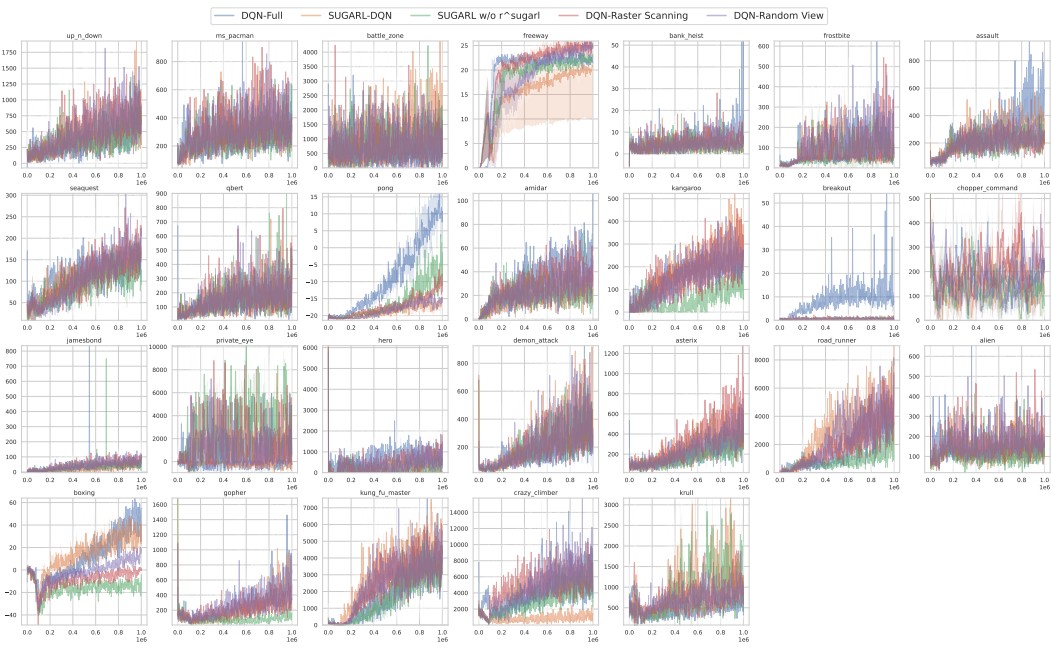

Figure 9: Learning curves of 26 Atari games, under the setting of 50x50 foveal observation size and 20x20 peripheral observation.

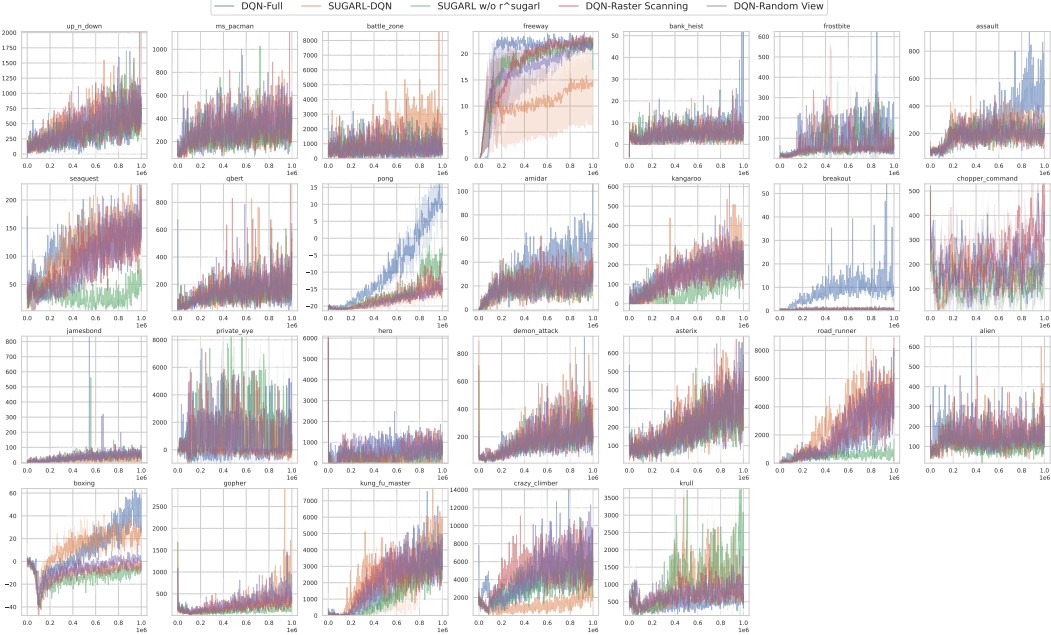

Figure 10: Learning curves of 26 Atari games, under the setting of 30x30 foveal observation size and 20x20 peripheral observation.

## A.2   Hyper-parameter Settings

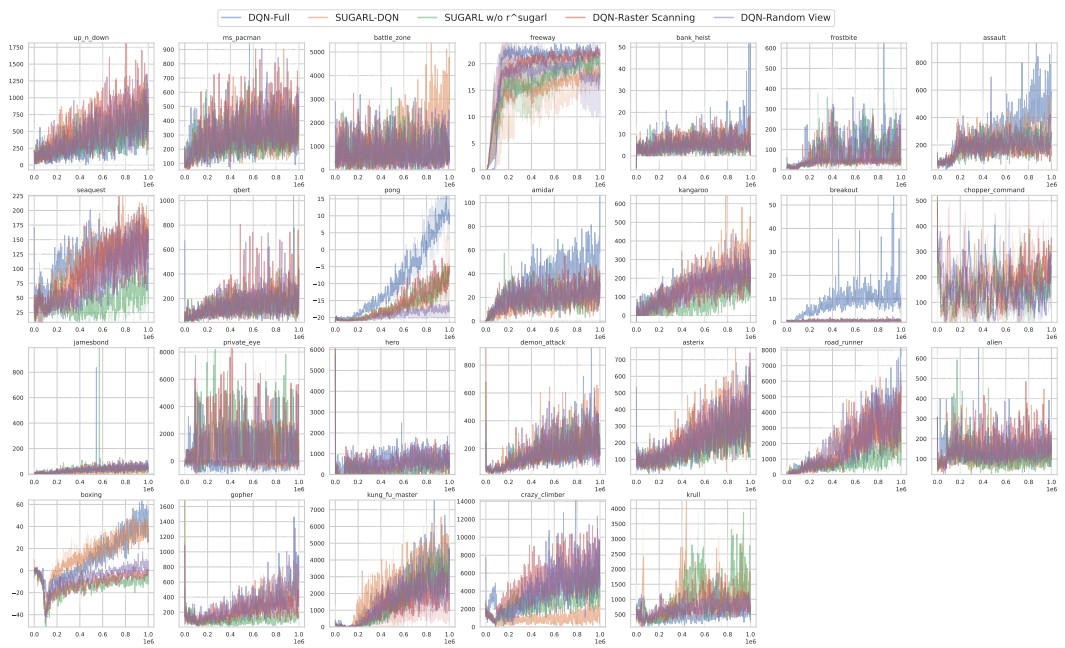

Figure 11: Learning curves of 26 Atari games, under the setting of 20x20 foveal observation size and 20x20 peripheral observation.

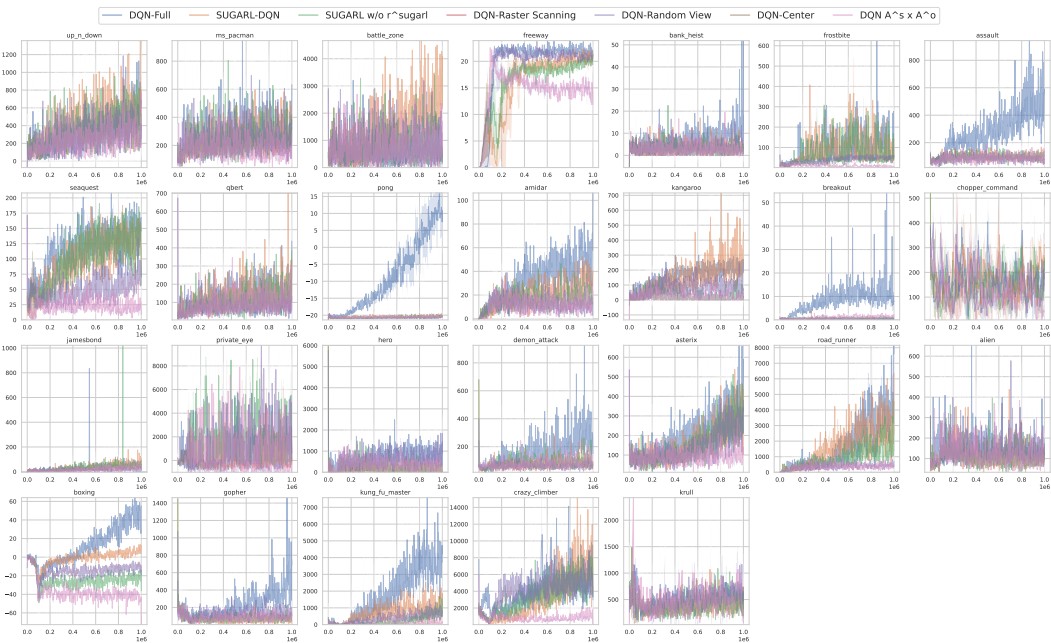

Figure 12: Learning curves of 26 Atari games, under the setting of 50x50 foveal observation size and w/o peripheral observation.

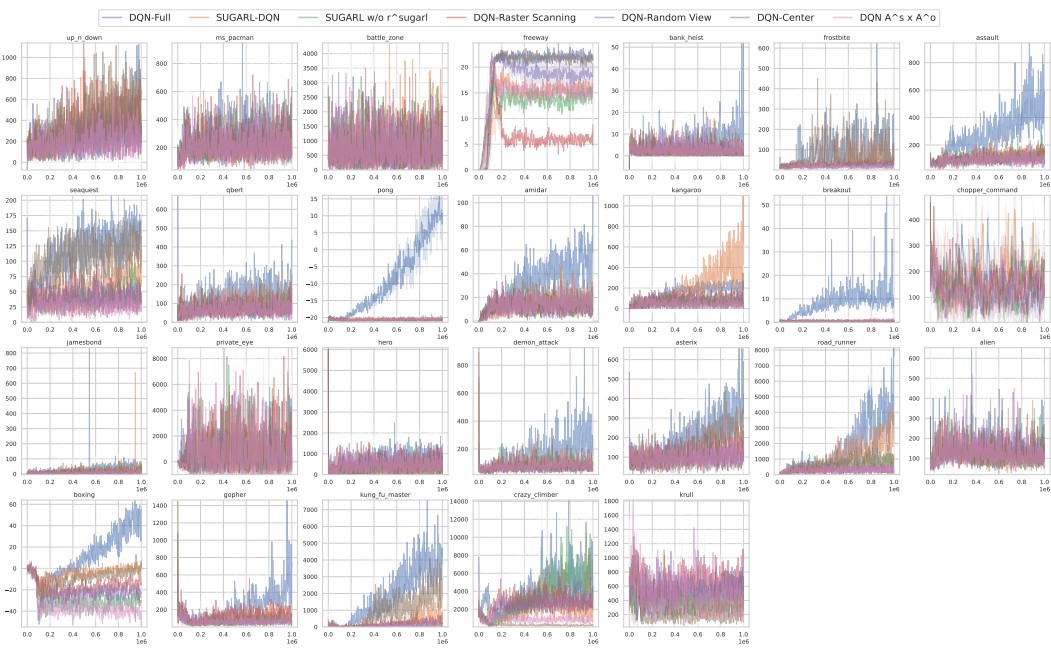

Figure 13: Learning curves of 26 Atari games, under the setting of 30x30 foveal observation size and w/o peripheral observation.

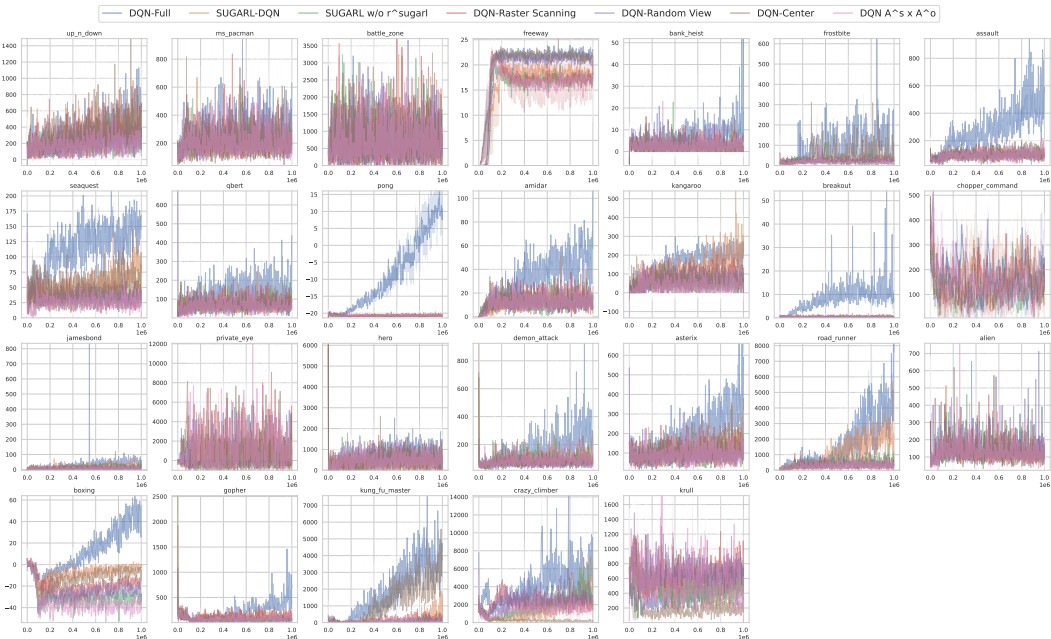

Figure 14: Learning curves of 26 Atari games, under the setting of 20x20 foveal observation size and w/o peripheral observation.

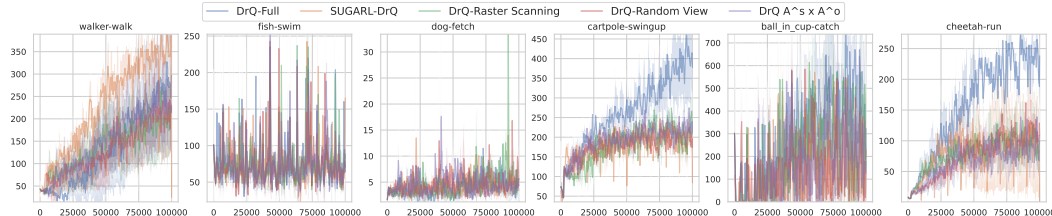

Figure 15: Learning curves of 6 DMC environments, under the setting of 50x50 foveal observation size and w/o peripheral observation.

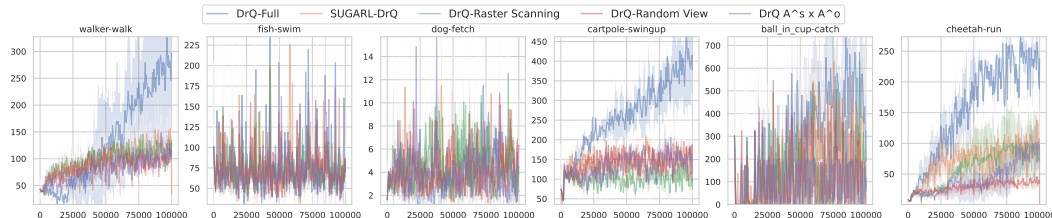

Figure 16: Learning curves of 6 DMC environments, under the setting of 30x30 foveal observation size and w/o peripheral observation.

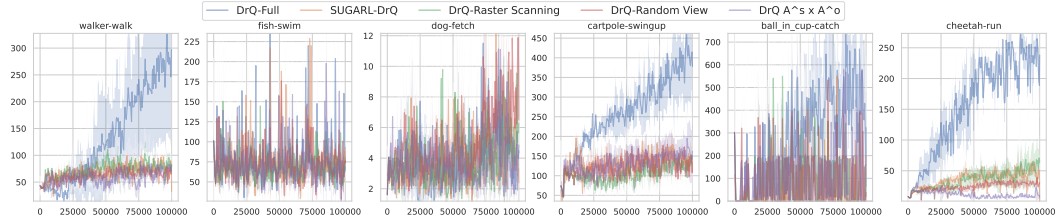

Figure 17: Learning curves of 6 DMC environments, under the setting of 20x20 foveal observation size and w/o peripheral observation.

Table 6: Hyper-parameters for DQN / SUGARL-DQN (on Atari)

| | |
|---:|---|
| Total steps | 1,000,000 or 5,000,000 |
| Replay buffer size | 100,000 |
| $\epsilon$ start | 1.0 |
| $\epsilon$ end | 0.01 |
| $\min \epsilon$ step | 100,000 |
| $\gamma$ | 0.99 |
| Learning start | 80,000 |
| Q network train frequency | 4 |
| Target network update frequency | 1,000 |
| Learning rate | $10^{-4}$ |
| Batch size | 32 |
| Self-understanding module train frequency | 4 |
| Self-understanding module learning rate | $10^{-4}$ |

Table 7: Hyper-parameters for SAC (on Atari)

| | |
|---:|:---|
| Total steps | 1,000,000 |
| Replay buffer size | 100,000 |
| $\gamma$ | 0.99 |
| Learning start | 80,000 |
| Actor train frequency | 4 |
| Critic train frequency | 4 |
| Target network update frequency | 8,000 |
| Actor Learning rate | $3 \times 10^{-4}$ |
| Critic Learning rate | $3 \times 10^{-4}$ |
| Batch size | 64 |
| Self-understanding module train frequency | 4 |
| Self-understanding module learning rate | $3 \times 10^{-4}$ |
| Visual policy alpha | 0.2 |
| Physical policy alpha | autotune |
| Physical policy target entropy scale | 0.2 |

Table 8: Hyper-parameters for DrQv2 (on DMC)

| | |
|---:|:---|
| Total steps | 100,000 |
| Replay buffer size | 100,000 |
| $\gamma$ | 0.99 |
| Standard deviation start | 1.0 |
| Standard deviation end | 0.1 |
| Standard deviation end step | 50,000 |
| Standard deviation clip | 0.3 |
| Learning start | 2,000 |
| Actor train frequency | 2 |
| Critic train frequency | 2 |
| Target network update frequency | 2 |
| Target network exponential moving average weight | 0.01 |
| Actor Learning rate | $10^{-4}$ |
| Critic Learning rate | $10^{-4}$ |
| Batch size | 256 |
| Self-understanding module train frequency | 2 |
| Self-understanding module learning rate | $10^{-4}$ |
| Multiple-step reward | 3 |

Table 9: Hyper-parameters for DrQv2 (on Robosuite)

| | |
|---:|:---|
| Total steps | 100,000 |
| Replay buffer size | 100,000 |
| $\gamma$ | 0.99 |
| Standard deviation start | 1.0 |
| Standard deviation end | 0.1 |
| Standard deviation end step | 50,000 |
| Standard deviation clip | 0.3 |
| Learning start | 8,000 |
| Actor train frequency | 2 |
| Critic train frequency | 2 |
| Target network update frequency | 2 |
| Target network exponential moving average weight | 0.01 |
| Actor Learning rate | $10^{-4}$ |
| Critic Learning rate | $10^{-4}$ |
| Batch size | 256 |
| Self-understanding module train frequency | 2 |
| Self-understanding module learning rate | $10^{-4}$ |
| Multiple-step reward | 3 |

Table 10: Environment Settings

| | |
|---:|:---|
| **Atari** | |
| Gray-scale | True |
| Full observation size | 84x84 |
| Frame stacking | 4 |
| Action repeat (frame skipping) | 4 |
| Observable area initial location | $(0,0)$ |
| Sensory action options | $4 \times 4$ grid |
| Sensory action space size | 16 (abs) or 5 (rel) |
| PVM number of steps | 3 |
| **DMC** | |
| Gray-scale | True |
| Full observation size | 84x84 |
| Frame stacking | 3 |
| Action repeat (frame skipping) | 2 |
| Observable area initial location | $(0,0)$ |
| Sensory action options | $4 \times 4$ grid |
| Sensory action space size | 5 (rel) |
| PVM number of steps | 3 |
| **Robosuite** | |
| Gray-scale | False |
| Full observation size | 84x84 |
| Frame stacking | 3 |
| Action repeat (frame skipping) | 2 |
| Observable area initial location | Side View |
| Sensory action space | continuous relative 5-DoF control |
| PVM number of steps | 3 |

