# OpenReview forum: "Active Vision Reinforcement Learning under Limited Visual Observability"
_NeurIPS.cc/2023/Conference — NeurIPS 2023 poster_

### Official Review · Reviewer_Rg6w · 2023-07-01

**Soundness:** 3 good
**Presentation:** 3 good
**Contribution:** 3 good
**Rating:** 5
**Confidence:** 3

**Summary:**

This paper studies an interesting setting where an RL agent needs to simultaneously decide how to act in the motor space and how to act to change the 2D observation space. Under this setting, the authors propose a new framework named SUGARL, mainly consisting of three technical contributions:
- a two-branch joint learning framework;
- a definition of the sensorimotor reward using inverse dynamics model
- a technique to accumulate history observations named Persistence-of-Vision Memory

The authors conduct experiments on Atari with DQN and on DMControl with DrQv2, to show the effectiveness of their framework compared to Random View and Raster Scanning baselines.

**Strengths:**

1. An interetsting problem named Active RL is reformulated and well-studied. This problem has its real-world meaning and could lead to possibly large impact for the real-world applications of Visual RL.
2. To drive for the better learning of the sensorimotor policy, a new intrinsic reward is defined based on IDM, which is well motivated and also effective in experiments.
3. The evaluation looks extensive, conducted on two kinds of environments with both continous and discrete action space, and every component is well ablated.
4. The learned sensorimotor policy gives relatively interesting visualizations, where the task-oriented objects are more focused.

**Weaknesses:**

1. **Environments are not matching the problem setting.** Although the proposed setting and the motivation example (i.e., robotic manipulation) are interesting to me, my major concern is the mismatch between the proposed setting and the experiment environments, where authors only focus on the 2D RL environments (DMC and Atari). Recent years have seen great advances in robotic simulation environments (e.g, MetaWorld [1], RoboSuite [2], ManiSkill [3], RLBench [4], ...), and it would be good to see the active RL setting is studied on these robotic environments, instead of the toy 2D environments such as video game.
2. **Baselines are weak.** The presented experiment results show that SUGARL outperforms Random View and Raster Scanning, which are two random weak baselines that could not show the advantage of the proposed framework. It would be good if authors could devise some stronger baseline (such as using a pre-trained detection network [5] to get the target, or using some hand-crafted features like color and shape to get the target) and compare.

[1] Yu, Tianhe, et al. "Meta-world: A benchmark and evaluation for multi-task and meta reinforcement learning." *Conference on robot learning*. PMLR, 2020.

[2] Zhu, Yuke, et al. "robosuite: A modular simulation framework and benchmark for robot learning." *arXiv preprint arXiv:2009.12293* (2020).

[3] Mu, Tongzhou, et al. "Maniskill: Generalizable manipulation skill benchmark with large-scale demonstrations." *arXiv preprint arXiv:2107.14483* (2021).

[4] James, Stephen, et al. "Rlbench: The robot learning benchmark & learning environment." *IEEE Robotics and Automation Letters* 5.2 (2020): 3019-3026.

[5] https://github.com/facebookresearch/detectron2

**Questions:**

Please see `weaknesses` for questions.

**Limitations:**

The limitations of the environments are not fully mentioned in the conclusion section, which however are my major concern in this work. Other parts are good.

---

> ### Author Rebuttal · Authors · 2023-08-08
>
> We thank the reviewer for recognizing our contributions. Please find our response below.
>
> **1\. Experiments on robotics tasks**
>
> Following the suggestion, we conducted new experiments on Robosuite [Zhu et al., 2020] and found that SUGARL performs best, especially on harder manipulator tasks. Please see our general response and Table 1 in our rebuttal PDF.
>
> **2\. Baselines**
>
> We follow the suggestion and introduce more SOTA baselines. Please also refer to our general response and Table 1 in our rebuttal PDF for details. We implement SOTA baselines including (a) DrQ with object detection using pre-trained DETR [Carion et al., 2020], and (b) DrQ with learned attention [Tang et al., 2020]. The former is our replication of a relevant approach from [Cheng et al., 2018], with a stronger object detection module and a stronger RL algorithm. We find that SUGARL outperforms both baselines by a large margin.
>
> **3\. Discussion on limitations**
>
> Following the suggestion, we provide a limitation section which we will incorporate to the revised paper.
>
> In this work, we assume that completely independent sensory and motor actions are present in an embodied agent. But in a real-world case, the movement of the sensors may depend on the motor actions. For example, a fixed camera attached to the end-effector of a robot manipulator, or to a mobility robot. To address the potential dependence and conflicts between two policies in this case, extensions like voting or weighing across two actions to decide the final action may be required. The proposed algorithm also assumes a chance to adjust viewpoints at every step. This could be challenging for applications where the operational or latency costs for adjusting the sensors are high like remote control. To resolve this, additional penalties on sensory action and larger memorization capability are potentially needed. Last, the intrinsic reward currently only considers the accuracy of agent-centric prediction. Other incentives like gathering novel information or prediction accuracy over other objects in the environment can be further explored.
>
> We appreciate your consideration of our clarifications.
>
> **References**
>
> - Carion et al., End-to-End Object Detection with Transformers, ECCV 2020
> - Tang et al., Neuroevolution of Self-Interpretable Agents
> - Cheng et al., Reinforcement Learning of Active Vision for Manipulating Objects under Occlusions, CoRL 2018
> - Zhu et al., robosuite: A modular simulation framework and benchmark for robot learning, 2020

---

> > ### Comment · Reviewer_Rg6w · 2023-08-11
> >
> > Thank the authors for providing new results. Here are my questions to the authors' replies.
> >
> > **Q1:**"Active RL agent is allowed to control the camera." Which camera is controlled?
> >
> > **Q2:** Could the authors show the visualization (e.g., videos about how the camera is changed by the policy) of the senor policy in robotic environments, like the videos originally provided?
> >
> > **Q3:** In the implemented stronger baseline, i.e., DrQ with DETR, is the bounding box provided as additional information, or the images cropped by the bounding box provided as additional information? I hope authors could explain more on the implementation details.

---

> > > ### Author Response · Authors · 2023-08-14
> > >
> > > Thanks for your reply!
> > >
> > > **1\. Which camera is controlled**
> > >
> > > We initialize the movable camera at the location of one of the hand-coded "side view" from robosuite. It is on the left of the robot.
> > > We also report the performance of standard RL using this "side view" in the below table. The performance is lower than standard RL in other hand-coded views (results in the rebuttal PDF), showing that learning from this view is challenging.
> > > |           | Wipe |  Door | NutAssenblySquare | Lift | Stack |
> > > |-----------|:----:|:-----:|:-----------------:|:----:|:-----:|
> > > | Side view | 25.9 | 136.2 |        34.5       | 56.6 |  12.8 |
> > >
> > > **2\. Visualization**
> > >
> > > Sure. Please check our video at https://drive.google.com/file/d/1Ay-ZdJMF3ekjNe1sa7Gt9fqMgtxb-itb/view?usp=sharing
> > >
> > >  **3\. DrQ with DETR baseline**
> > >
> > > We provide the 3D positions of the detected objects, inferred from bounding boxes (from DETR) and camera pose (from the environment), replicating the design in [Cheng et al., 2018].

---

> > > > ### Author Response · Authors · 2023-08-19
> > > >
> > > > Dear Reviewer Rg6w,
> > > >
> > > > We appreciate your taking the time to review our paper and respond to our rebuttal. We have provided the clarifications and new materials requested. It will be great to let us know if your concerns are addressed.
> > > >
> > > > Thanks,
> > > > The Authors

---

> > > > > ### Comment · Reviewer_Rg6w · 2023-08-19
> > > > > **A good start!**
> > > > >
> > > > > Thank the authors for providing the new results in a 3D robotic environment. I appreciate the efforts made by the authors.
> > > > >
> > > > > However, the new experiments on robotic tasks are just *a good start*, and compared to the extensive experiments conducted in the initial paper, these new results are not sufficient to make it a NeurIPS paper. Also, the new visualization is showing the proposed method is not working as well in 2D environments.
> > > > >
> > > > > Same as other reviewers, I hope the authors could make large changes to their original paper, such as extensive robotic environments, new illustrations on their method and implementation (since the current paper only focuses on 2D toy environments), and maybe improvements over their old method, to enhance it under the robotic environments.
> > > > >
> > > > > Therefore, after carefully considering all the results, I would change my score to reject.

---

> > > > > > ### Author Response · Authors · 2023-08-20
> > > > > >
> > > > > > Thank you for reviewing the paper. However, we believe there are misunderstandings going on.
> > > > > >
> > > > > > > the new visualization is showing the proposed method is not working as well in 2D environments
> > > > > >
> > > > > > We provided quantitative results (Table 1 of the rebuttal pdf) clearly illustrating that SUGARL is even more effective in the 3D environment than 2D. The relative performance gain we got with SUGARL (over the standard single policy baseline) in 3D is 5-10x as shown in Table 1 of the rebuttal pdf (74.1 vs. 12.4 for Wipe, 291.0 vs. 22.8 for Door, 65.2 vs. 8.42 for NutAssemblySquare, …). The gain we had is 3-5x in 2D Atari as shown in Table 1(a) of the paper. The agent is able to benefit from our algorithm, and we provided the quantitative results to support the argument.
> > > > > >
> > > > > > > compared to the extensive experiments conducted in the initial paper, these new results are not sufficient
> > > > > >
> > > > > > Our robosuite experiments are shown in Table 1 of the rebuttal pdf. We believe they are as extensive as the main experiments we have done in the paper. We evaluated 5 distinct robot manipulation tasks with 3 different seeds. We compared SUGARL variants against 9 baselines:
> > > > > > - These include SOTA baselines using object detection and using end-to-end attention, which we newly added.
> > > > > > - Similar to the experiments with Atari, we compared them against 4 hand-coded viewpoint baselines and 1 single policy baseline.
> > > > > > - We investigated SUGARL with multiple versions of PVM, as well as provided ablations with/without our joint learning and PVM.
> > > > > >
> > > > > > > Same as other reviewers, I hope the authors could make...
> > > > > >
> > > > > > Which reviewers do you mean? Other than one reviewer who has not given any feedback to the rebuttal, all three reviewers recognized the new experiment and clarifications.
> > > > > >
> > > > > > > extensive robotic environments
> > > > > >
> > > > > > We have already provided them in the rebuttal pdf.
> > > > > >
> > > > > > > new illustrations on their method and implementation
> > > > > >
> > > > > > We have the new PVM described in the rebuttal, and will be provided further in the final version of the paper. Other than that there is no change needed to apply SUGARL on 3D tasks.
> > > > > >
> > > > > > > improvements over their old method, to enhance it under the robotic environments
> > > > > >
> > > > > > We also provided them in Table 1 of the rebuttal PDF. Row 1 vs. 2-4 in the table corresponds to this.

---

> > > > > > > ### Comment · Reviewer_Rg6w · 2023-08-21
> > > > > > >
> > > > > > > Thank the authors for the explanation. This summarization addresses my concerns.
> > > > > > >
> > > > > > > I would raise my score accordingly.
> > > > > > >
> > > > > > > Moreover, I hope the code for both 2D and 3D environments could be released, especially the robotic environments, which could be potentially beneficial for the community.

---

### Official Review · Reviewer_3Naa · 2023-07-04

**Soundness:** 3 good
**Presentation:** 4 excellent
**Contribution:** 3 good
**Rating:** 7
**Confidence:** 5

**Summary:**

The paper introduces an RL setting where an agent take actions to simultaneously perform a task and control its vision. The authors design an algorithm for learning two policies for task performing and vision controlling. They test the algorithm on two domains: Atari and DMC. Results show that the proposed algorithm works better than baselines that do not intentionally control the vision. The authors conduct a comprehensive study to analyze the behavior of the algorithm.

**Strengths:**

* The proposed setting is interesting. While it is not clear in the paper, I can envision potential significant impact.
* The paper is very well-written. The methods are clearly presented and the results are thoroughly analyzed.
* The experiment analyses are thoughtful and comprehensive. The behavior of the algorithm is dissected extensively, bringing valuable insights to the readers.

**Weaknesses:**

My concerns are about the high-level framing of the paper and the lack of comparison with highly relevant baselines:
* The motivation of the setting is still unconvincing in terms of practicability. Scientifically, it is interesting to replicate human abilities to intentionally adapt their field of vision, but it is unclear why acquiring the same abilities would be pragmatically beneficial for an AI agent, compared with (i) observing all information or (ii) end-to-end-learned attentions.
   * The paper seems to be missing a notion of "cost" (e.g., time budget, cognitive effort, amount of information), which makes it easier to motivate and compare different methods. For example, in figure 5(b), the method outperforms observing the full state, but it is unclear whether the gain is due to observing more pixels in general or having the ability to control the vision. A more fair comparison would fix a budget of pixels (or some other type of cost) for all methods.
   *  The authors should include baselines with end-to-end-learned attention. Without any cost of gathering new information, one can design a baseline that scans the full state, and applies attention on top of that. With this baseline, I am currently not convinced why learning attention through RL would be more effective than through end-to-end optimization.
* I suggest changing the framework to "Active*Vision*-RL", which fits better to the scope of the paper.


**Questions:**

I do not have any questions.

**Limitations:**

There is no limitations section. I suggest discussing the limitations of the simulated environments in modeling real-world scenarios, the drawbacks of RL in the face of rich information and action spaces.

---

> ### Author Rebuttal · Authors · 2023-08-08
>
> We thank the reviewer for recognizing our contributions. We will address the concerns below.
>
> **1\. The necessity of active perception**
>
> We thank the reviewer for the constructive comments. In order to highlight the necessity of active perception and effectiveness of our approach.
>   - We ran new experiments on Robosuite [Zhu et al., 2020] where an active perception is naturally beneficial for manipulation. Our results show that active agents are outperforming hand-coded views most of the time. Please refer to our general response for the details of the experiments.
>   - We also compared our proposed approach with baselines with (i) observing all information (using an object detector) or (ii) end-to-end-learned attentions. SUGARL outperforms them. Please also refer to Tables 1 and 3 in our rebuttal PDF for detailed results. For the attention-based baseline, we notice there are several possible approaches for RL like [James et al., 2022, Tang et al., 2020, Wu et al., 2021]. We choose [Tang et al., 2020] because it is differentiable and with only a small amount of computation overhead for fair comparison.
>
>
>
> **2\. Pixel budget**
>
> Please see the table below for comparison on the number of observed pixels versus performance under Atari. In Atari and DMC, the number of actual pixels observed by the full observation model is 84x84=7056. For the foveal only settings, SUGARL uses 3x20x20=1200 (17.0%), 3x30x30=2700 (38.3%), 3x50x50=7500 (106%) pixels **at most**, where 3 is the introduced PVM length, and 20, 30, 50 are the foveal resolutions. We note that there are overlaps (because of stitching) between old and new observations, so the actual numbers will be smaller than those, especially for the 50x50 case. SUGARL models have lower pixel budgets than the full observation model in general. In Robosuite, all methods have the same pixel budget. We will add these comparisons to the paper.
>
> | Visual Settings                 | # Observed Pixels   | Percentage of a Full Obs. | Performance on Atari |
> |---------------------------------|---------------------|:--------------------------:|:----------------------:|
> | 20x20 foveal only               | <20x20x3=1200       | <17.0%                   | 0.475                |
> | 30x30 foveal only               | <30x30x3=2700       | <38.3%                   | 0.810                |
> | 50x50 foveal only               | <50x50x3=7500       | <106%                    | 0.805                |
> | 20x20 foveal + 20x20 peripheral | <20x20x3+20x20=1600 | <22.7%                   | 1.010                |
> | 30x30 foveal + 20x20 peripheral | <30x30x3+20x20=3100 | <43.9%                   | 1.116                |
> | 50x50 foveal + 20x20 peripheral | <50x50x3+20x20=7900 | <112%                    | 1.289                |
> | Full                            | 84x84=7056          | 100%                     | 1.000                |
>
> **3\. Term ActiveVision-RL**
>
> Thanks for the suggestion, we will follow it.
>
> **References**
>
> - James et al., Q-attention: Enabling Efficient Learning for Vision-based Robotic Manipulation, RA-L 2022
> - Tang et al., Neuroevolution of Self-Interpretable Agents, GECCO 2020
> - Wu et al., Self-Supervised Attention-Aware Reinforcement Learning, AAAI 2021
> - Zhu et al., robosuite: A modular simulation framework and benchmark for robot learning, 2020

---

> > ### Comment · Reviewer_3Naa · 2023-08-11
> > **Thanks for your repsonse**
> >
> > I appreciate the additional experiments. I decide to raise my score to 7. I hope the authors will incorporate the rebuttal materials into the next version.

---

### Official Review · Reviewer_Wwp7 · 2023-07-06

**Soundness:** 2 fair
**Presentation:** 3 good
**Contribution:** 3 good
**Rating:** 6
**Confidence:** 5

**Summary:**

The authors develop an RL framework for problems where the agent must control its perception in order to solve the task. Their framework, SUGARL, decouples the task into a sensory and motory policy, and propose a heuristic reward for training the sensory policy stably. They evaluate in Active Vision versions of Atari and DMC.

---
(8/14/23) The rebuttal has addressed my major concern, which is showing SUGARL results in more realistic robotics tasks in 3d scenes. I also appreciate the additional baselines the authors added in.

Updating my score from 3 to a 6.

I would recommend the authors to focus their time now on making the paper easier to read, as well as performing experiments on two more realistic robotic settings - a setting where moving the camera has a cost, and a setting where there are physical occluders.



**Strengths:**

- An RL framework that sensibly handles active perception by decoupling the overall task into sensory and motor tasks.
	- The authors propose a heuristic dense reward for the sensory policy that seems to work well in their tasks, although I have some questions about its generality. Nevertheless, I appreciate its simplicity and effectiveness.
- Interesting and diverse experiments, although experiments are in rather artificial active vision settings (see concerns).
	- Sometimes, Active-RL can even outperform Full Observation baseline with joint training of the active vision policy.  This is impressive, since training both active vision and motor policies from scratch can easily destabilize the RL agent.
	- I like the qualitative analysis of the sensory policy behavior.

**Weaknesses:**

## Major
**Experimental section does not evaluate active perception in natural settings.**
 - There is an obvious omission of tasks in natural domains where active perception is necessary. All tasks are currently done in a modified version of Atari or DMC where the agent can instantly change the view per timestep. In more realistic settings, such as robotics, changing the sensors have costs - energy cost, delays, etc. These constraints are not mentioned or studied at all.
 - Neurips is a more theoretical conference, but this paper focuses on solving Active RL problems, which is mainly a robotics problem. So while the Atari/DMC benchmark adds some scientific value, the experimental section needs a more realistic Active RL benchmark to show practical value. The authors should show results on a robotics task, like manipulation or navigation as done in related work [1,2]. I am also open to discussing this point, but the authors would need to convince me that the Atari/DMC tasks are somehow reflective of actual Active-RL applications.

This is the most pressing issue for me, and I believe if addressed, will make this paper very compelling. If SUGARL can show good trends (outperform or competitive with full view baseline) in realistic applications, it will be a good method since joint training the sensory and motor policies together is nontrivial.

## Moderate
**Experiment section is messy.** It has the feeling of being cobbled together rather than presenting a unified analysis. There are many small experiments and their figures are all over the place. This makes it hard for the reader to understand key takeaways. For example, section 5.1 seems to have 5 "mini" experiments, each asking a different question. Some experiments seem more tangential (e.g. training for more steps) and can be moved to appendix.  While it's hard for me to tell exactly what needs to be fixed here, overall this section needs to be rewritten so the reader can easily get key takeaways. Can the authors attempt to make this section easier to read?


**PVM is an implementation trick for 2D environments rather than a contribution.** Related to my major concern on using only artificial 2D environments - the authors propose Persistence of Vision memory, which stitches together multiple recent images into one. This is more of an implementation trick for 2D environments. Can we give baselines access to PVM, or remove PVM from SUGARL for comparison? I see there is already an ablation for PVM, but I would like to see this across all tasks. Alternatively, the authors can evaluate in a 3D environment (like robotics, navigation, etc.) and show a 3D version of PVM. Either way would make it more general.

**SUGARL reward on non agent-centric tasks.**  The SUGARL reward rewards the sensory policy for choosing views with low action prediction error. However, will this still work in tasks where relevant parts of the image are unrelated to the action? For example, Pong and Pinball have moving small balls that are important to look at, but the balls give little information about the action.

**Reproducibility.** No codebase or promise of code release of the Active-Gym benchmark or SUGARL algorithm.

1. Cheng, Ricson, Arpit Agarwal, and Katerina Fragkiadaki. "Reinforcement learning of active vision for manipulating objects under occlusions." _Conference on Robot Learning_. PMLR, 2018.
2. Wang, Hanqing, et al. "Active visual information gathering for vision-language navigation." Computer Vision–ECCV 2020: 16th European Conference, Glasgow, UK, August 23–28, 2020, Proceedings, Part XXII 16. Springer International Publishing, 2020.

**Questions:**

## Questions from the limitations:
Can the authors provide results on a more realistic benchmark where active perception is required, like manipulation or benchmark?

Can the authors provide a more "fair" comparison with baselines by either removing or adding PVM?

Does the SUGARL reward work in environments where action prediction is not helpful for solving the task?


## Clarification questions:

It says that both policies are trained with a shared reward function. Does this mean that the motor policy can also optimize the sugarl reward? If so, couldn't this lead to degenerate case where motor policy only moves in very predictable ways?

**Limitations:**

The authors say one sentence for limitations - that Active-RL learns more slowly than full observations. This is fairly obvious, and I would like to see more discussion on other limitations. I can suggest some.

This framework makes many assumptions about the active perception itself. First, it assumes a clean separation between motor and sensory actions - they do not affect each other. However, in many tasks, *interactive perception* is required - a robot may choose to move its arm away from the camera or open a box to remove occlusion. These actions can count as both motor and sensory actions. Another assumption is that the active perception actions happen instantly alongside motor actions, and have no cost. This is not true in the real world. Can SUGARL handles these assumptions, or what extensions would be necessary to address these assumptions?

Next, what are the compute and resource constraints of training an Active-RL policy versus a normal policy? Can the authors provide more details about GPU usage, runtime of experiments, etc.?

---

> ### Author Rebuttal · Authors · 2023-08-08
>
> We thank the reviewer for recognizing the effectiveness of our approach and providing thoughtful comments. Following the suggestions, we conducted addtional experiments on more realistic robot environments.
>
> **1\. Active perception settings**
>
> Following the suggestion, we tested our algorithm on a more realistic robot control (simulated) environment, Robosuite [Zhu et al., 2020]. In this environment, the camera motion is controlled by 5-DoF control: relative (x, y, z, yaw, pitch), and is constrained by maximum linear and angular velocities. Under this constrained active perception setting, we confirm that SUGARL works the best.
>
> In the Atari results we provided, we also test SUGARL with relative control (SUGARL(rel), Table 1(a) in the paper). This control only allows the view to move to nearby locations. From the experimental result we observe that it sometimes can outperform absolute control.
>
> **2\. Robotics task**
>
> We followed the suggestion and conducted new experiments on Robosuite. Results are available in the general response and Table 1 in the rebuttal PDF. In short, SUGARL achieves best performance on 3 of 5 challenging manipulation tasks (are all harder tasks), and outperforms hand-coded views most of the time on all tasks.
>
> **3\. Organization of experiment section**
>
> Thanks for the comment, we will improve this section accordingly. For example, having a list of questions we try to answer at the beginning of this section.
>
> **4\. Contribution of PVM**
>
> Please see our general response (2. Designs of PVMs). We conducted extra experiments on Robosuite where we make several new instantiations of PVM for this 3D case. The results are in Table 1 of the rebuttal PDF. We find the version for the 3D environment is effective in combining partial observations spatially and temporally. PVM is about finding an appropriate function to combine partial observations spatially and temporally, and we provide several versions of in these new experiments.
>
> Following the suggestion, we tested SUGARL w/o PVM on Robosuite and DMC (Tables 1, 2 in the rebuttal PDF). We confirm that PVM is an important component.
>
> **5\. Non-agent centric tasks/sensory policy behavior**
>
> The sensory policy is learned and will focus on the necessary part. It is not forced to pay attention to specific parts like surroundings. In the meantime, the presence of environmental reward encourages both policies to be task-oriented. So the sensory action won't overly satisfy the intrinsic reward.
>
> One example of this is the visualization of Assault (the middle of Row 2) in the supplementary website available in the caption of Figure 6. The sensory policy focuses on the enemies (top) rather than the agent (bottom). In the visualization of Pong (Row 7), the sensory policy observes the ball many times.
>
> **6\. Effects of shared reward function**
>
> The final reward we use is the combination of environmental reward $r^{env}$ and our intrinsic reward $r^{sugarl}$. With the presence of environmental reward, the motor policy won’t overly satisfy intrinsic reward. The reward balance is the key and can be determined in a principled way from previous research [Hasselt et al., 2016, Henderson et al., 2018, Choi et al., 2019]. We verify its effectiveness and robustness across different visual settings in Atari and DMC. We didn’t observe a highly predictable motor policy from our current study.
>
> **7\. Discussion on limitations**
>
> Following the suggestion, we provide a limitation section which we will incorporate to the revised paper.
>
> In this work, we assume that completely independent sensory and motor actions are present in an embodied agent. But in a real-world case, the movement of the sensors may depend on the motor actions. For example, a fixed camera attached to the end-effector of a robot manipulator, or to a mobility robot. To address the potential dependence and conflicts between two policies in this case, extensions like voting or weighing across two actions to decide the final action may be required. The proposed algorithm also assumes a chance to adjust viewpoints at every step. This could be challenging for applications where the operational or latency costs for adjusting the sensors are high like remote control. To resolve this, additional penalties on sensory action and larger memorization capability are potentially needed. Last, the intrinsic reward currently only considers the accuracy of agent-centric prediction. Other incentives like gathering novel information or prediction accuracy over other objects in the environment can be further explored.
>
> **8\. Reproducibility and resources**
>
> We will open-source our approach and the benchmark, including the robotics environment.
> The resources required are listed below, measured using a NVIDIA A5000 GPU. The speed is mostly bottlenecked by the underlying simulator (if only one thread is used), not our algorithm.
> | Environment | Algorithm    | Steps | RAM | GPU VRAM | Wall-clock Time/Task |
> |-------------|--------------|-------|-----|----------|----------------------|
> | Atari       | DQN          | 1M    | 18G | 1.3G     | 1.2 hrs              |
> | Atari       | SUGARL-DQN   | 1M    | 18G | 1.7G     | 1.6 hrs              |
> | DMC         | DrQv2        | 100K  | 18G | 2.5G     | 1.4 hrs              |
> | DMC         | SUGARL-DrQv2 | 100K  | 18G | 2.8G     | 1.7 hrs              |
> | Robosuite   | DrQv2        | 100K  | 54G | 3.9G     | 2.5 hrs              |
> | Robosuite   | SUGARL-DrQv2 | 100K  | 54G | 4.2G     | 3 hrs                |
>
> Thank you for reading our responses.
>
> **References**
> - Henderson et al., Deep reinforcement learning that matters, AAAI 2018
> - Hasselt et al., Learning values across many orders of magnitude, NeurIPS 2016
> - Choi et al., Intrinsic motivation driven intuitive physics learning using deep reinforcement learning with intrinsic reward normalization, 2019
> - Zhu et al., robosuite: A modular simulation framework and benchmark for robot learning, 2020

---

> > ### Comment · Reviewer_Wwp7 · 2023-08-12
> > **Good start, hope to see some more progress.**
> >
> > Thank you for the initial rebuttal, I think it is a good start for addressing my concerns. Updated score from 3 to 4 to reflect the improvement so far.
> >
> > ## More realistic robotics experiments
> > I appreciate the new robotics experiments. However, they are still not complete - analysis of the experiments is missing. I would like to see some more analysis of them, like videos, comparisons of behavior, and why does Active-RL work or not work in certain tasks compared to the baselines.
> >
> > I don't get the "Adv over hand coded views" column, can you give more details about this metric? Is it the number of evaluation episodes where one method gets higher returns than another?
> >
> > ## PVM concern is addressed.
> > The authors show how to use PVM for 3d images, by proposing multiple ways of fusing pixel information over time. This is quite interesting itself, and addresses my concern about PVM.
> >
> > ## Bonus Experiment on Cost of Perception
> > Can you design a robotics task where perception has a cost? This is a more realistic setup. For example, you can put some energy cost for moving the camera too much. It would be interesting to see how sensory policies learn to efficiently perceive the scene as well.

---

> > > ### Author Response · Authors · 2023-08-14
> > >
> > > We thank the reviewer for the positive feedback. We have the visualization video showing the learned policies at https://drive.google.com/file/d/1Ay-ZdJMF3ekjNe1sa7Gt9fqMgtxb-itb/view?usp=sharing
> > >
> > > **Explanations on robotics experiments**
> > >
> > > > Policy behavior
> > >
> > > We observe that the learned sensory policy exhibits multiple active-vision skills like tracking the end-of-effector, moving to focus on task-oriented objects, or moving to focus on the whole scene. Details are available in the accompanying video.
> > >
> > > > why does Active-RL work or not work in certain tasks compared to the baselines
> > >
> > > Compared to learning from hand-coded views, we find that SUGARL works better for harder tasks. The movable camera view could potentially gather more information that helps motor policy. In those easier tasks including Lift and Stack, the moving camera may cause unnecessary difficulty in observing the target and the status of the gripper.
> > >
> > > > Adv over hand coded views
> > >
> > > Sorry for the confusion. For each method other than hand-coded views, we compare it to each of three hand-coded views in each of the five tasks. We count one for having a higher IQM than a hand-coded view in one task. So the maximum is 15 if one method beats all hand-coded views in all tasks. This is to measure how an active-vision agent performs compared to standard RL under the hand-coded view.

---

> > > > ### Comment · Reviewer_Wwp7 · 2023-08-14
> > > >
> > > > Thanks for the clarification and analysis of the robotics experiments. It is quite interesting that ActiveRL can improve these robotics tasks, even when there is no obvious occlusion. This is quite compelling since it means a lot of visual manipulation tasks can benefit from active vision even if their cameras were set up to be fully observed.
> > > >
> > > > My last remaining concern is that there are no learning curves of the robotics experiments - could you provide those as well?
> > > >
> > > > Once addressed, I have no immediate concerns about this paper. I do have some further suggestions to improve it though:
> > > >
> > > > 1) The cost of perception experiment that I mentioned earlier.
> > > > 2)  Adding occluding obstacles into the environment, and seeing if ActiveRL can look around the occlusion.

---

> > > > > ### Author Response · Authors · 2023-08-14
> > > > >
> > > > > Please find the learning curves at https://drive.google.com/file/d/1mnR1Ce-g6xvNfi6y2VPei3EWwjTHJLSr/view?usp=sharing.
> > > > >
> > > > > Thanks for the suggestions on further improvements and we will follow them.

---

> > > > > > ### Comment · Reviewer_Wwp7 · 2023-08-14
> > > > > > **Updating my score from 3 to a 6.**
> > > > > >
> > > > > > I would recommend the authors to focus their time now on making the paper easier to read, as well as performing experiments on two more realistic robotic settings - a setting where moving the camera has a cost, and a setting where there are physical occluders.

---

> > > > > > > ### Author Response · Authors · 2023-08-19
> > > > > > >
> > > > > > > Thank you for the nice discussion and suggestions. We will modify our paper accordingly, and try our best to investigate the suggested setting.

---

### Official Review · Reviewer_SANf · 2023-07-06

**Soundness:** 3 good
**Presentation:** 2 fair
**Contribution:** 2 fair
**Rating:** 7
**Confidence:** 4

**Summary:**

This paper presents a method for what the authors call "active RL", a flavor of RL where the agent not only needs to pick a (motor) action, but also needs to select the (sensory) action where to look. The authors do so by training a DQN or SAC agent with separate heads for the motor and sensory policies respectively. The sensory actions are learned using a separate reward function, which is based on the prediction error when predicting the motor action from two consecutive observations. The authors evaluate on Atari and DMC equipped with limited observability.

**Strengths:**

- Active vision is an important skill that we humans use all the time, but is often overlooked in the RL literature. It's nice to see an "active RL" approach addressing this issue.

- The paper is well written, and presents nice quantitative and qualitative results on both DMC and Atari.

**Weaknesses:**

- I found the very coarse discretization of sensory actions limiting.

- The method seems very dependent on the reward balance hyperparameter.

**Questions:**

- In the case no PVM is used, to you still provide the complete frame size, but with the unseen pixels black-ed out. Would it also work when just providing the foveal observation?

- The sugarl reward is focused on predicting the agent's motor action. However, in many games you should rather pay attention to the surroundings, since you "know" your action (i.e. especially when standing still for instance), but you'd want to know if there are new obstacles popping up at the edges. Have you also tried other incentives, for instance expected information gain, which supposedly also underpins human saccading, i.e.
https://www.frontiersin.org/articles/10.3389/fncom.2016.00056/full
https://www.frontiersin.org/articles/10.3389/fnbot.2022.840658/full

- It's interesting that SUGARRL outperforms vanilla RL with full observations on some environments. Do you have any insights on why this might be the case? Is there anything common on the particular environments where the gap is largest?

- Would it help to provide the policy with the current sensory action, i.e. so it knows where it is currently looking at. Although it probably won't make a difference in case the PVM / full observation with black pixels is used, as it can derive it from that. Still, it feels like a waste of compute to process all these black pixels.

- Could the sensory policy be generalized across environments?

**Limitations:**

The main limitation mentioned is the slower training. I'd say the limited set of fixed fixation points is also a limitation of the current implementation. It would be nice to scale this to an agent that e.g. has a continuous pan/tilt control for its camera.

---

> ### Author Rebuttal · Authors · 2023-08-08
>
> We thank the reviewer for recognizing our technical contribution and our analysis. Please find our response below.
>
> **1\. Sensory action space**
>
> We follow the suggestion to test SUGARL on continuous camera control on Robosuite, a simulated robotics environment. The agent uses a 5-DoF relative (x, y, z, yaw, pitch) to control the camera. Our results are available in general response and Table 1 in rebuttal PDF. We find that SUGARL is also effective in continuous sensory action setting for challenging robotics tasks.
>
> **2\. Reward balance**
>
> Similar to other research [Hasselt et al., Henderson et al., Choi et al.], the reward balance hyperparameter is necessary to make rewards at different scales to work. We follow those previous research to select this balance. The hyperparameter is computed as the max/average environmental return normalized by the length of the trajectory. In our experiments on Atari and DMC, we use only one hyperparameter per game/task for all visual settings. SUGARL performs well consistently. And this hyperparameter strategy also works for our new Robosuite experiments. These verify the robustness of balance selection.
>
> **3\. Pixel utilization / foveal observation without black padding**
>
> We find out that providing the foveal observation without padding (IQM=0.400) is not better than the approach with PVM (IQM=0.805). Providing those blacked pixels is helping the performance.
>
>
> **4\. Sensory policy incentives and behaviors**
>
> Thanks for these inspirational references. We will look into the connections between ours and the references, and will investigate new forms of incentives in the future.
>
> The sensory policy is learned and will focus on the necessary part. It is not forced to pay attention to specific parts like surroundings. The presence of environmental reward encourages both policies to be task-oriented. In this case, the sensory action won't overly satisfy the intrinsic reward. Furthermore, due to the existence of PVM, the agent is able to access observation several steps before. This allows the agent to use information gathered from different locations, and thus supports the potential "saccading" skills.
>
> **5\. Insights on outperforming some full observation baselines**
>
> Our insight is that the partially observable area excludes noisy information that may not be helpful for decision making.
> For example, the “scores” either on the top or the bottom of the game screen can be noisy. They are changes in pixels, but they do not mean the change of actual “state” like object locations.
>
> Battle_zone is actually one of them with the largest performance improvements than the full observation model (>3x). As shown in Figure 6, the “fixation” behavior makes visual observation fixed to the center of the screen, which is the only direction the agent can fire a bullet. This "simplified" observation allows easier policy learning. More visualization is available in the caption of Figure 6.
>
> **6\. Providing sensory action location as input**
>
> Thanks for the thoughtful question! PVM indeed implies the sensory action location. And it is able to precisely combine observations spatially. We tried to provide such information to the policy but it does not further improve.
>
>
> **7\. Sensory policy transfer**
>
> This work is under the single-task setting, so it’s hard to measure the transferability of the learned agent. In visual RL, the transfer of the visual encoder itself could be challenging before studying the transfer of the policy [Hansen et al.], as it has more parameters. To make it work, multi-task (pre-)training might be needed and it is a completely different setting from this paper. The transfer of learned policy is a valuable direction and we will leave it for future work. Thanks for the valuable question!
>
> **8\. Continuous camera control**
>
> We follow the suggestion and conduct new experiments on Robosuite, with continuous control of the camera. According to the results in Table 1 in the general rebuttal PDF, SUGARL works the best. Please also refer to our general response for more details.
>
> Thank you for reading our responses.
>
> **References**:
>
> - Henderson et al., Deep reinforcement learning that matters, AAAI 2018
> - Hasselt et al., Learning values across many orders of magnitude, NeurIPS 2016
> - Choi et al., Intrinsic motivation driven intuitive physics learning using deep reinforcement learning with intrinsic reward normalization, 2019
> - Hansen et al., Stabilizing Deep Q-Learning with ConvNets and Vision Transformers under Data Augmentation, NeurIPS 2021

---

> > ### Comment · Reviewer_SANf · 2023-08-11
> >
> > I thank the authors for their detailed responses to my questions. I especially appreciate the new Robosuite experiment, and increased my score accordingly.

---

### Official Review · Reviewer_LTdh · 2023-07-07

**Soundness:** 2 fair
**Presentation:** 3 good
**Contribution:** 2 fair
**Rating:** 5
**Confidence:** 3

**Summary:**

This work focuses on reinforcement learning with a controllable perceptive field where the agent not only learns a motor policy but learns a sensor policy at the same time. The sensor policy controls the information to be obtained. This work proposes to coordinate two policies by introducing an intrinsic reward encouraging the sensor policy to provide observation correlates to the decision making. The proposed method is evaluated on modified DMC and atari benchmarks to show the effectiveness of the proposed method.

**Strengths:**

1. The proposed method is intuitive and easy to implement, enough technical details are included for future reproducement.
2. This work performed extensive ablation study over different components of the proposed method and showed effectiveness of different components.
3. This work provides clear visualization of learned sensor policy for better understanding of the work, and discusses the behavior of the sensor policy under different observation resolutions.
4. The overall writing of the work is good, the paper is easy to follow and easy to understand.

**Weaknesses:**

1. The problem itself is not novel, previous work[1] has studied similar settings with more challenging tasks, separately modeling the sensor policy and the motor policy should not be considered a contribution as well. With only proposed new intrinsic reward functions, the novelty of the work is relatively limited.
2. The proposed Persistence-of-Vision Memory module is specifically designed for the given task, which can hardly apply to more realistic tasks like moving cameras with different poses.
3. According to the detailed experiment results in supplementary materials, the experiment results are super noisy and the performance varies a lot according to the environment.
4. Some baselines like DrQ / DQN with separate heads for motor policy and sensory policy should be compared to validate the method. These baselines should not be considered as part of the ablation study.

[1] Cheng, et al; Reinforcement Learning of Active Vision for Manipulating Objects under Occlusions

**Questions:**

1. How does DrQ or DQN with separate heads for sensor policy and motor policy work in different benchmarks?
2. How does other forms of memory work in this case, like LSTM which is much more generalizable to other applications compared with the proposed PVM module.

**Limitations:**

Potential negative societal impact is not applicable to this work. For POMDP, providing some kind of memory mechanism for agent to learn a better policy is quite standard practice in the community, some baseline with LSTM or other memory mechanism should be compared.

---

> ### Author Rebuttal · Authors · 2023-08-08
>
> Thanks for the comments and questions. Please see below for our response that addresses the concerns.
>
> **1\. Novelty of this research**
>
> We would like to highlight our contributions as follows:
>
> - The decomposition of policies governing sensory and motor actions. It’s a different approach compared to conventional single-policy approaches including [Cheng et al., 2018]. We experimentally show that ours works much better (Table 1 in the rebuttal PDF) on difficult Robosuite manipulator tasks.
> - The new approach to enable learning of such decomposed policies, based on joint learning with intrinsic reward. And we empirically show the joint learning approach is easy to implement by simple extensions from many existing RL algorithms, and it works well.
> - We bring up the concept of PVM and show different instantiations of it. We confirm their effectiveness through experiments in 2D and 3D environments.
> - We provide comprehensive analysis on sensory policy behavior that helps understanding and is beneficial for further sensory policy / joint learning algorithm design.
>
> And another key difference is that [Cheng et al., 2018] was originally designed for goal-conditioned RL, which has access to the privileged information.
>
> **2\. More instantiations of Persistence-of-Vision Memory (PVM)**
>
> The Persistence-of-Vision Memory is proposed to be a general framework that combines multiple recent partial observations into one using a function $f()$. What we present in the paper is one implementation of it (i.e., 2D-image stitching), highlighting its necessity. Following the suggestion, we further provide more instantiations of $f()$. We additionally implemented:
>
> - **LSTM-based PVM**: Each image is first encoded by CNN and then fed into LSTM
> - **3D Transformation + Stacking**: We use camera parameters to align pixels from different images to the current camera frame. Then the transformed images are simply stacked on the channel axis.
> - **3D Transformation + LSTM**: Similar 3D transformation as above, but using an LSTM to encode the image sequence.
>
> The results in Table 1 in rebuttal PDF show that they can handle 3D environments. In particular, the approach of using both temporal LSTM and 3D alignment for PVM is most effective. It can effectively handle the camera motion and improve policy learning.
>
> The motivation behind the PVM design is to combine visual observations from multiple **spatio-temporal** locations. We study its influence on the challenging active vision + RL problems. While recurrent models like LSTMs could provide a memorization capability to a certain degree, explicit spatial-temporal encoding (e.g., 2D stitching or 3D Transformation) further improves the memory.
>
>
> **3\. Performance and learning curves**
>
> The performance varies across environments due to different environment properties, which is very common in RL algorithms. For example, in Appendix E of DreamerV2 [Hafner et al., 2021], Figure 3 of SimPle [Kaiser et al., 2020], the performance also varies a lot.
>
> For the curves, we show the raw episodic return during training without smoothing. They are noisy due to random explorations in the learning process. We will improve the visualization to make it clear.
>
> **4\. More comparison using separate heads**
>
> Following this suggestion, we provide more results of DrQ with separate heads (i.e. **SUGARL-DrQ w/o Joint Learning**) in Robosuite (Table 1 in the rebuttal PDF) and in DMC (Table 2 in the rebuttal PDF). Both experiments show separated heads are not producing better results than full SUGARL.
>
> Thank you for reading our responses.
>
> **References**
>
> - Cheng et al., Reinforcement Learning of Active Vision for Manipulating Objects under Occlusions, CoRL 2018
> - Hafner et al., Mastering Atari with Discrete World Models, ICLR 2021
> - Kaiser et al., Model Based Reinforcement Learning for Atari, ICLR 2020

---

> > ### Comment · Reviewer_LTdh · 2023-08-20
> >
> > After reading the response, I'd like to raise my score to borderline accept. The new instantiations of PVM look reasonable, the authors provide some new results in the response PDF, it would be great if the authors could include all new results in the modified version

---

### Author Rebuttal · Authors · 2023-08-08

**General response:**

We thank all reviewers for inspiring comments and questions. In this general response, we address the common concerns of more experiments on (1) robot manipulator tasks, (2) more baselines, and (3) more designs of proposed Persistence-of-Vision Memory (PVM).

1. **New experiments on robot manipulator tasks**

    In addition to Atari and DMC, following the suggestions from the reviewers, we test SUGARL on Robosuite [Zhu et al., 2020], a simulated robotics environment.

    **Tasks**: We selected five of available tasks, namely block lifting (Lift), block stacking (Stack), nut assembling (NutAssembleSquare), door opening (Door), wiping the table (Wipe). The first two are easier compared to the later three. Example observations are available in Figure 1 in rebuttal PDF.

    **Camera / sensory action setting**:
    Active RL agent is allowed to control the camera. We use a 5-DoF **continuous** control: relative (x, y, z, yaw, pitch). The maximum linear and angular velocities are constrained to 0.05/step and 5 degrees/step, respectively. For reference, the dimension of the table is 0.8x0.8.

    **Results**:
    We compare baselines including RL with object detection (a replication of [Cheng et al., 2018]), learned attention [Tang et al., 2020], and standard RL with hand-coded views. Results are in Table 1 in the new rebuttal PDF. We confirm that our SUGARL works outperforms SOTA baselines all the time, and also outperforms the hand-coded views most of the time. Specifically, for the harder tasks including Wipe, Door, and NutAssemblySquare, SUGARL gets the best scores.

2. **More SOTA baselines**

    Following the reviewer suggestion, we implement two SOTA baselines.
    - **DrQ w/ Object Detection**: Since [Cheng et al., 2018] requires goal-conditioning unlike our setting, we extend [Cheng et al., 2018] with a stronger object detector and RL algorithm. We try our best to replicate [Cheng et al., 2020] using a pre-trained object detector DETR [Carion et al., 2020] that provides object bounding boxes in addition to visual observation. Compared to the original version of [Cheng et al., 2018], the reimplemented method has (a) a stronger detector DETR and (b) DrQv2 which is an improved version of DDPG. We remove the goal conditioning in [Cheng et al., 2018].
    - **DrQ w/ End-to-End Attention**: We use [Tang et al., 2020] to perform an learnable attention. This approach divides the input image to patches and selects K patches with the highest attention scores for the policy input. We use it along with DrQv2 in order to perform a fair comparison.

    From the experimental results in Table 1 in the rebuttal PDF, we find that SUGARL outperforms them by a large margin. They perform similarly or slightly better than the Single Policy baseline. In Table 3 in the rebuttal PDF, we also find that SUGARL outperforms attention approach in Atari.

3. **Designs of PVMs**

    The Persistence-of-Vision Memory (PVM) is proposed to be a general framework, and what we present in the paper is one implementation of it (i.e., 2D-image stitching). Following the suggestion of the reviewers, we further provide more instantiations of it. We additionally implemented it using
    - **LSTM**: Each image is first encoded by CNN and fed into LSTM.
    - **3D Transformation + Stacking**: We use camera parameters to align pixels from different images to the current camera frame. Then the transformed images are simply stacked on the channel axis. CNN encodes the stacked images.
    - **3D Transformation + LSTM**: Similar 3D transformation as above, but using an LSTM to encode the images after going through CNN.

    We compare these new instantiations to a simple stacking PVM in Robosuite, and results are available in the first 4 rows of Table 1 in rebuttal PDF. We find that 3D Transformation + LSTM works the best, because it tackles spatial aligning and temporal merging together. LSTM also works well in general.

    We also test LSTM PVM on Atari to compare it with our originally proposed Stitching PVM. The results are available in Table 3 of the rebuttal PDF. We find that LSTM PVM does not outperform Stitching PVM because Stitching can combine observations exactly according to their spatial locations.


**Details of robosuite experiments**

  - **Hand-coded view baselines:** We select three hand-coded views that come with Robosuite -- **Front View**, **Agent View**, and **Eye-in-hand View**. These are used for the standard RL. Front View has the camera facing towards the robot. Agent View has the camera closer to the robot than Front View and more concentrated to the working area (e.g. table top or the door). The Eye-in-hand View is from the camera attached to the end effector of the robot. Front View and Agent View are static views while the Eye-in-hand View camera moves together with the end effector. Examples are available in Figure 1 in rebuttal PDF.

  - **Training:** Each task is trained for 100k steps. The size of the replay buffer is 100k. Each task is trained for 3 seeds, with 10 evaluation runs at the end of the training.

  - **Metrics:** We report the absolute IQM value from a total of 30 evaluations.

Thank you for reading our responses.

**References**
- Carion et al., End-to-End Object Detection with Transformers, ECCV 2020
- Cheng et al., Reinforcement Learning of Active Vision for Manipulating Objects under Occlusions, CoRL 2018
- Tang et al., Neuroevolution of Self-Interpretable Agents, GECCO 2020
- Zhu et al., robosuite: A modular simulation framework and benchmark for robot learning, 2020

---

### Decision · Program_Chairs · 2023-09-21

**Decision:**

Accept (poster)

**Comment:**

This paper proposes a method for active reinforcement learning where two control policies are learned, one for motor actions and one for sensor actions. The motor and sensor policies are learned jointly by optimizing a combination of environmental and intrinsic rewards. In initial reviews, the primary concerned shared by the reviewers was  the evaluation tasks used. The authors addressed this concern in their rebuttal by performing additional experiments on Robotsuite. I encourage the authors to include these experiments in the final version as their inclusion increases the impact and value of the paper.